# An evaluation of SMOS L-Band vegetation optical depth (L-VOD) data sets: a high sensitivity of L-VOD to above-ground biomass in Africa

Nemesio J. Rodríguez-Fernández[1], Arnaud Mialon[1], Stephane Mermoz[1], Alexandre Bouvet[1], Philippe Richaume[1], Ahmad Al Bitar[1], Amen Al-Yaari[2], Martin Brandt[3], Thomas Kaminski[4], Thuy Le Toan[1], Yann H. Kerr[1], and Jean-Pierre Wigneron[2]

[1]Centre d'Etudes Spatiales de la Biosphère (CESBIO), Université de Toulouse, Centre National d'Etudes Spatiales (CNES), Centre National de la Recherche Scientifique (CNRS), Institut de Recherche pour le Dévelopement (IRD), Université Paul Sabatier, 18 av. Edouard Belin, bpi 2801, 31401 Toulouse, France
[2]Interactions Sol Plante Atmosphére (ISPA), Unité Mixte de Recherche 1391, Institut National de la Recherche Agronomique (INRA), CS 20032, 33882 Villenave d'Ornon cedex, France
[3]Department of Geosciences and Natural Resources Management, University of Copenhagen, 1350 Copenhagen, Denmark
[4]The inversion Lab, Hamburg, Germany

*Correspondence to:* N.J. Rodríguez-Fernández (nemesio.rodriguez@cesbio.cnes.fr)

**Abstract.** The vegetation optical depth (VOD) measured at microwave frequencies is related to the vegetation water content and provides information complementary to visible/infra-red vegetation indices. This study is devoted to the characterisation of a new VOD data set obtained from SMOS (Soil Moisture and Ocean Salinity) satellite observations at L-band (1.4 GHz). Three different SMOS L-band VOD (L-VOD) data sets (SMOS Level 2, Level 3 and SMOS-IC) were compared with data sets on tree height, visible/infra-red indexes (NDVI, EVI), mean annual precipitation, and above ground biomass (AGB) for the African continent. For all relationships, SMOS-IC showed the lowest dispersion and highest correlation. Overall, we found a strong ($R > 0.85$) correlation with no clear sign of saturation between L-VOD and four AGB data sets. **The relationships between L-VOD and the AGB data sets were linear per land cover class, but with a changing slope depending on the class type, which makes a global non-linear relationship. In contrast, the relationship linking L-VOD to tree height ($R = 0.87$) was close to linear. For vegetation classes other than evergreen broadleaf forest,** the annual mean of L-VOD spans a range from 0 to 0.7 and it is linearly correlated with the amount of the average annual precipitations. SMOS L-VOD showed a higher sensitivity to AGB as compared to NDVI and K/X/C-VOD (VOD measured, respectively, at 19, 10.7, and 6.9 GHz). The results showed that although the spatial resolution of L-VOD is coarse ($\sim 40$ km), the high temporal frequency and sensitivity to AGB makes SMOS L-VOD a very promising **indicator** for large scale monitoring of the vegetation status, in particular biomass.

## 1 Introduction

Large scale monitoring of vegetation properties is crucial to understand water, carbon and energy cycles. The Normalized Difference Vegetation Index (NDVI, Tucker, 1979) computed from space-borne observations at visible and infra-red wavelengths

has been widely used since the 1980s to study vegetation changes and its implications on animal ecology (Pettorelli et al., 2005, 2011), global fire emissions (Van der Werf et al., 2010), deforestation and urban development (Esau et al., 2016), global patterns of land-atmosphere carbon fluxes (Jung et al., 2011) and the vegetation response to climate (Herrmann et al., 2005) and extreme events such as droughts (Vicente-Serrano et al., 2013). NDVI is sensitive to the abundance of chlorophyll and

therefore to the photosynthetically active biomass (which includes herbaceous vegetation and the leaves of trees), but insensitive to wood mass. NDVI is thus not considered as an accurate proxy of total above ground biomass (AGB), except in areas of low vegetation density (Todd et al., 1998). Contrastingly, being sensitive to both green and non-green vegetation components, passive microwave observations can provide important complementary information on the state and time changes of the vegetation features, in particular regarding the AGB dynamics (Liu et al., 2015).

The thermal emission arising from the Earth surface at microwave frequencies depends on the soil characteristics such as soil temperature, soil roughness and soil moisture content, which controls the soil emissivity (Ulaby, 1976). In the presence of vegetation, part of the soil emission is absorbed and scattered. **These effects can be parameterized using radiative transfer models such as the so-called $\tau - \omega$ model (Mo et al., 1982; Ulaby and Wilson, 1985; Ferrazzoli and Guerriero, 1996; Wigneron et al., 2007; Liu et al., 2011), where $\tau$ is the optical depth and $\omega$ is the single scattering albedo.** $\tau$ was shown

to be linked to the vegetation water content (VWC, kg/m$^2$) (Kirdiashev et al., 1979; Mo et al., 1982; Jackson and Schmugge, 1991) and to other vegetation properties such as the Leaf Area Index (Jackson and Schmugge, 1991; Van de Griend and Wigneron, 2004; Wigneron et al., 2007). **Therefore, $\tau$ is commonly known as Vegetation Optical Depth (VOD).** VOD is also a function of the vegetation structure which determines its dependence on the incidence angle and on the polarization of the radiation (Ulaby and Wilson, 1985; Wigneron et al., 1995, 2004; Hornbuckle et al., 2003; Schwank et al., 2005).

Passive microwave radiometry is therefore a promising tool to monitor the vegetation at global scale. VOD samples the vegetation canopy including woody vegetation, which uses root zone soil moisture (Andela et al., 2013). VOD was used to study deforestation in South America (Van Marle et al., 2016) and Africa (Brandt et al., 2017). Using VOD, it has been possible to reveal teleconnexions linking the state of the vegetation in Australia and El Niño Southern Oscillation (Liu et al., 2007). In addition, Liu et al. (2015) showed the high potential of microwave VOD to monitor the AGB dynamics at large scale. Using

both VOD and NDVI contributes to provide a more robust assessment of the vegetation characteristics (Liu et al., 2011). **The VOD has also been used to study the VWC and variations in ecosystem-scale isohydricity (Konings and Gentine, 2017; Li et al., 2017).**

The above mentioned studies used VOD derived from different radiometers operating at different frequencies (Liu et al., 2011): SSM/I at 19 GHz (K-band), TRMM-TMI at 10.7 GHz (X-band), and the Advanced Microwave Scanning Radiometer -

Earth Observing System (AMSR-E) at 10.7 GHz and 6.9 GHz (C-band). It is worth noting that VOD is intrinsically dependent on the frequency of the electromagnetic radiation and VODs retrieved at different frequencies provide complementary information. Therefore, in the following, a specific VOD data set will be noted as *B*-VOD, where *B* stands for the microwave band (X-VOD, C-VOD,...). The lower the frequency, the lower the VOD for a given level of VWC (Wigneron et al., 1995, 2004; Ferrazzoli and Guerriero, 1996). Consequently, L-band (1.4 GHz, 21 cm) observations, which are less attenuated through the

vegetation canopy, are capable of sampling the vegetation layer up to higher biomass values compared to higher frequency observations.

Currently, two missions are performing systematic L-band passive microwave observations: The Soil Moisture and Ocean Salinity (SMOS) satellite (Kerr et al., 2010), launched by ESA in November 2009, and the Soil Moisture Active Passive (SMAP) satellite (Entekhabi et al., 2010), launched by NASA in January 2015. **SMAP measures the brightness temperature for a single incidence angle in two polarizations. A single-angle dual polarization retrieval algorithm decreases the quality of the soil moisture retrievals (Konings et al., 2016) but using a multi-orbit approach assuming that the L-VOD does not vary significantly in a few days window, it is possible to estimate soil moisture and L-VOD (Konings et al., 2017)**. The full-polarization and multi-angular capabilities of SMOS allow to retrieve simultaneously the soil moisture content and L-VOD. Lawrence et al. (2014) and Grant et al. (2016) compared SMOS L-VOD to X-VOD and C-VOD measured by AMSR-E and to visible/infra-red vegetation indices. In crop zones, as the MODIS vegetation indices, L-VOD increases during the growing season and decreases during senescence (Lawrence et al., 2014). At global scale, L-VOD is less correlated to optical/visible vegetation indices than X/C-VOD, suggesting that L-VOD can add more complementary information with respect to optical/infrared indices than X/C-VOD (Grant et al., 2016). For instance, Rahmoune et al. (2014) found a significant correlation between L-VOD and tree height estimates. Vittucci et al. (2016) also discussed this relationship and compared it to the one estimated with X/C-VOD, which shows higher values for low tree-height than SMOS L-VOD, as expected. Vittucci et al. (2016) also showed a close to linear relationship between L-VOD and AGB at 20 selected points over Peru, Columbia, and Panama. **L-VOD has been recently used to study the evolution of carbon stocks in African drylands by Brandt et al. (2018).**

In summary, L-VOD derived from the new SMOS L-Band observations is a promising tool for monitoring global vegetation characteristics. There is, however, a lack of in-depth studies on how L-VOD relates to established vegetation characteristics. The goal of the current study is to get further insight into the sensitivity of L-VOD to vegetation properties **(such as tree height and AGB)** and precipitations, which can drive the vegetation dynamics for some biomes. Taking into account the novelty of these observations, three distinct SMOS L-VOD data sets were evaluated against several data sets independent of L-VOD: (*i*) optical/infra-red indices (representing the greenness of vegetation, also often used as proxy for primary productivity), (*ii*) AGB benchmark maps, (*iii*) LIDAR derived tree height and (*iv*) precipitation data set. The area selected for this study is Africa, as it is a continent with several climate regions and biomes and with a large variability in the vegetation biomass from sparse shrubs to savannah and very dense rainforests. In addition, Bouvet et al. (2018) have recently discussed the first biomass map of African savannahs computed from L-band active microwave (synthetic aperture radar) observations. **In contrast to passive measurements, for which the goal is to study how the thermal emission arising from the Earth is affected by the vegetation layer, active measurements allow to study how the radiation emitted by a human-made radiation source is backscattered by the vegetation, which depends mainly of the vegetation water content and the vegetation structure.**

**Since this study is mainly devoted to AGB, long time averages (typically annual) will be used. Studying the evolution of VWC would require using much shorter time scales.** The document is organized as follows. Section 2 presents the different SMOS L-VOD data sets as well as the data sets used for the evaluation (tree height, cumulated precipitations, NDVI,

EVI and four AGB data sets). Section 3 deals with the evaluation methods. Section 4 presents the results, which are discussed in Section 5, in particular the potential of L-VOD to estimate AGB at large scale. Finally, Section 6 summarizes the results and presents the conclusions of this study.

## 2 Data

### 2.1 SMOS data

The SMOS (Kerr et al., 2001, 2010) mission is an ESA-led mission with contributions from CNES (Centre National d'Etudes Spatiales, France) and CDTI (Centro Para el Desarrollo Tecnológico Industrial, Spain). The SMOS radiometer measures the thermal emission from the Earth in the protected frequency range around 1.4 GHz in full-polarization and for incidence angles from $0°$ to $\sim 60°$. **Stokes 3 and 4 parameters are used to filter the data, to detect radio frequency interference sources, for instance.** The footprint (full width at half maximum of the synthesized beam) is $\sim 43$ km on average (Kerr et al., 2010). The equator overpass time is 6:00 AM/PM for ascending/descending orbits. **Ascending and descending orbits data from 2011 and 2012 are used in this study.** Taking into account the novelty of L-VOD estimates, three different L-VOD data sets were evaluated in this study: the ESA Level 2 (L2) product, the CATDS multi-orbit Level 3 (L3) product and the new INRA-CESBIO (IC) data set (Table S1 gives a summary of the main characteristics of those three products).

The three SMOS soil moisture and L-VOD L2 retrieval algorithms discussed below use the L-MEB (L-band Microwave Emission of the Biosphere) radiative transfer model (Wigneron et al., 2007), which is based on the $\tau - \omega$ parametrization to take into account the effect of vegetation. The soil temperature profile is estimated from European Centre for Medium Range Weather Forecasts (ECMWF) Integrated Forecast System (IFS) data. The difference between forward model estimates of the brightness temperatures at antenna reference frame and actual satellite measurements is minimized by varying the values of the soil moisture (SM) content and the L-VOD. The contributions from the soil and vegetation layers can be distinguished thanks to the multi-angular and dual-polarization measurements.

The differences between the three SMOS data sets are discussed in the following.

#### 2.1.1 SMOS Level 2 soil moisture and L-VOD

The SMOS soil moisture and L-VOD L2 retrieval algorithm was described by Kerr et al. (2012). The forward model contributions are computed at $\sim 4$ km resolution pixels and aggregated to the sensor resolution using the mean synthetic antenna pattern. For footprints with mixed land cover, the L2 algorithm distinguishes the minor and the major land cover (low vegetation or forest). The SMOS retrieval is performed only over the dominant land cover class within the footprint while the emission of the minor land cover is estimated from ECMWF SM and MODIS Leaf Area Index (LAI) data (Kerr et al., 2012). The version of the data used in the current study is 620. This data version uses auxiliary files including information on L-VOD computed from previous retrievals, surface roughness and Radio Frequency Interference (RFI) that are used to constrain the new retrievals. **Due to the specificities of the SMOS geometry of observation, the profiles of brightness temperatures**

observed at the middle part of the field of view (∼600 km centered on the satellite sub-track) have larger ranges of incidence angles than the outer part of the field of view. For such observations, the retrieval system has more information content to discriminate the vegetation emission from the ground emission leading to more accurate retrieved soil moisture and VOD. The retrieved VODs and associated uncertainties for such grid points are used as prior first guess and uncertainties for the L-VOD retrieval of the next overpass of these grid-points (3 days later maximum) that will be observed, this time, at the outer part of the field of view with a reduced range of incidence angle. This avoids to use auxiliary LAI data to compute a first-guess L-VOD value (Kerr et al., 2012).

The SMOS L2 data are provided by ESA in an Icosahedral Equal Area (ISEA) 4H9 grid (Sahr et al., 2003) in swath mode with a sampling resolution of 15 km. **The single scattering albedo and roughness values depend on the surface type and are taken from literature and/or specific SMOS studies. For low vegetation areas, the single scattering albedo is set to 0 and roughness set to 0.1. For forested areas the single scattering albedo is set to 0.06 for tropical and subtropical forest and 0.08 for Boreal forest and roughness set to 0.3 (Rahmoune et al., 2013, 2014).**

### 2.1.2 SMOS Level 3 soil moisture and L-VOD

The SMOS L3 soil moisture and L-VOD data set is provided by the CATDS (Centre Aval de Traitement de Données SMOS) from CNES (Centre National D'Etudes Spatiales) and IFREMER (Institut Français de Recherche pour l'Exploitation de la Mer) in an Equal-Area Scalable Earth (EASE) grid version 2 (Brodzik et al., 2012.) with a sampling resolution of 25 km. The data version used in this study is Version 300. The data set and the retrieval algorithm are described in Al Bitar et al. (2017). The L3 algorithm is based on the same physics and modelling as the ESA L2 single-orbit algorithm (Section 2.1.1). However, instead of using information on prior retrievals to constrain the SM and L-VOD inversion, the Level 3 algorithm uses a multi-orbit approach with data from three different revisits over a seven day window. In contrast to soil moisture, L-VOD is not expected to change strongly over a short period of time. Therefore a Gaussian correlation function is used during the retrieval to penalize large L-VOD variations in the cost function. The standard deviation of the Gaussian correlation function is 21 days for forests and 7 days for low vegetation. **The single scattering albedo and roughness parametrizations use the same approach and values of the L2 algorithm.**

### 2.1.3 SMOS INRA-CESBIO (IC) soil moisture and L-VOD

The SMOS INRA-CESBIO (SMOS-IC) algorithm was designed by INRA (Institut National de la Recherche Agronomique) and is produced by CESBIO (Centre d'Etudes Spatiales de la BIOsphère). A detailed description is given in Fernandez-Moran et al. (2017). One of the main goals of the SMOS-IC product is to be as independent as possible from auxiliary data, which are often also used for evaluation. In contrast to the L2 and L3 algorithms, the IC algorithm considers the footprints to be homogeneous to avoid uncertainties and errors linked to possible inconsistencies in the auxiliary data sets which are used to characterize the footprint heterogeneity. In addition, SMOS-IC differs from the SMOS L2 and L3 products in the initialization of the cost function minimization and in the modelling of heterogeneous pixels: no LAI nor ECMWF SM data are used. **A first run was done with SM 0.2 m³/m³ and L-VOD 0.5 as initial guess for the minimization. This allowed to compute**

**a mean L-VOD map per each grid point. The final inversion was done using this mean L-VOD map as first guess for L-VOD and a value of 0.2 m$^3$/m$^3$ as first guess for SM. The roughness and single scattering parameters are assigned per International Geosphere-Biosphere Program (IGBP, Loveland et al., 2000) land cover classes, based on Parrens et al. (2017b, a), and are averaged within a footprint according to the fraction of classes present in the footprint.** The data used in this study is version 103 and it is provided in the 25 km EASEv2 grid.

## 2.2 Evaluation data sets

This study performs an evaluation of the SMOS L-VOD data sets by a comparison with other vegetation-related evaluation data sets which are described in the following.

### 2.2.1 Precipitations

The Worldclim data set (Fick and Hijmans, 2017) provides spatially interpolated monthly climate data for global land areas at a very high spatial resolution (approximately 1 km). It includes monthly temperature (minimum, maximum and average), precipitation, solar radiation, vapour pressure and wind speed, aggregated across a target temporal range of 1970-2000, using data from between 9000 and 60 000 weather stations. As precipitations drive the vegetation dynamics for some biomes, mean annual precipitation were used to evaluate the relationship with L-VOD.

### 2.2.2 MODIS vegetation indices

MODIS NDVI and Enhanced Vegetation Index (EVI) from the product MYD13C1 (Tucker, 1979; Huete et al., 2002) collection 6 were compared to the SMOS L-VOD data sets to test L-VOD's performance against green photosynthetically active vegetation. Both NDVI and EVI are directly linked to the essential climate variables FAPAR and LAI and they are widely used as proxy for green vegetation cover. The NDVI product contains atmospherically corrected bi-directional surface reflectances masked for water, clouds, and cloud shadows.

EVI uses the blue band to remove residual atmospheric contaminations caused by smoke and sub-pixel thin cirrus clouds, which also introduces uncertainties over tropical areas. EVI was designed to have a higher sensitivity in high biomass regions than NDVI by allowing to distinguish the vegetation and the atmosphere contributions to the signal (Huete et al., 2002). Whereas the NDVI is chlorophyll sensitive, the EVI is more responsive to the canopy type and structure (including LAI) and, for example, it has allowed to study the Amazon green-up season (where other vegetation indexes such as NDVI do not show any particular pattern, Huete et al. (2006)).

Global MYD13C1 data are cloud-free spatial composites of the gridded 16-day 1 km MYD13A2, and are provided as a Level 3 product projected on a 0.05° geographic Climate Modeling Grid (CMG). Cloud-free global coverage is achieved by replacing clouds with the historical MODIS time series climatology record.

### 2.2.3 Lidar tree height

This study used global tree height data from Simard et al. (2011). This data set was produced using 2005 data from the Geoscience Laser Altimeter System (GLAS) aboard ICESat (Ice, Cloud, and land Elevation Satellite). The processing follows three steps. First, Simard et al. (2011) developed a procedure to select waveforms and correct slope-induced distortions and to calibrate canopy height estimates using field measurements. In a second step, GLAS canopy height estimations were found to be correlated to other ancillary data such as annual mean precipitation, precipitation seasonality, annual mean temperature, temperature seasonality, elevation, tree cover and classes of protection status. In a third step, a machine learning approach (random forest) was trained using the ancillary variables as input and GLAS tree height as reference data. Finally, the random forest algorithm was applied to the ancillary data to produce a forest canopy height map at 1 km resolution for areas not covered by GLAS waveforms.

### 2.2.4 Above ground biomass

This study used four static AGB benchmark maps (Baccini et al., 2012; Saatchi et al., 2011; Avitabile et al., 2016; Bouvet et al., 2018) each with specific strengths and limitations to assess L-VOD's ability to reflect aboveground biomass in different biomes: Whereas the maps produced by Saatchi, Baccini and Avitabile aim at covering all pan-tropical region, with focus on dense forests, the Bouvet's map focuses on African savannahs with lower biomass values. To take advantage of ALOS/PALSAR L-band observations, in the current study the Bouvet data set has also been extended to rainforest (see below).

The first AGB map over Africa was extracted from the 1 km resolution pan-tropical AGB data set produced by Saatchi et al. (2011). The methodology to produce this data set involves roughly two steps:

*(i)* in situ inventory plots are used to derive AGB estimates from the Lorey's height (the basal area weighted height of all trees with a diameter of more than 10 cm) calculated from the ICESat GLAS measurements,

*(ii)* these punctual measurements are spatially extrapolated using MODIS and Quick Scatterometer (QuikSCAT) data through Maximum Entropy (MaxEnt) modeling. All in situ AGB measurements were made from year 1995 to year 2005, and the MODIS and QuikSCAT data used for spatial extrapolation were acquired in 2000-2001, so that the resulting biomass map is representative of AGB circa the year 2000.

This study also used data over Africa extracted from the pan-tropical AGB data set produced by Baccini et al. (2012). The methodology used to produce this data set is very similar to that of Saatchi et al. (2011), except that *(i)* only MODIS data are used for the spatial extrapolation, *(ii)* Random Forest is used instead of MaxEnt, *(iii)* the data set is representative of circa 2007-2008, and *(iv)* the AGB map is produced at a resolution of 500 m.

The Avitabile et al. (2016) was also used in this study. This forest biomass data set was obtained by merging the data sets by Saatchi et al. (2011) and Baccini et al. (2012) with machine learning techniques to compute a pan-tropical AGB map at 1-km spatial resolution. The merging method was trained using an independent reference data set with field observations and locally calibrated high-resolution biomass maps, harmonized and up-scaled to be representative of 1 km$^2$. They used a total of 14477

AGB samples in Australia, Southern Asia, Africa, South America and Central America, spanning AGB values from 0 to $\sim 500$ Mg/h and covering different biomes such as grasslands, shrublands, savannahs and rainforests.

The fourth biomass map used in this study is based on Bouvet et al. (2018) map over savannahs and from Mermoz et al. (2015) over dense forests. The map from Bouvet et al. (2018) at 25 meter resolution is the first biomass map for Africa with focus on savannahs and was built from a L-band ALOS PALSAR mosaic produced with observations made in year 2010 (when SMOS was already in operation). A direct model was developed to relate the PALSAR backscatter to AGB with the help of in situ and ancillary data. In a subsequent step, a Bayesian inversion of the direct model was performed. Seasonal effects were taken into account by stratification into wet/dry season areas. In Bouvet et al. (2018), the method was originally applied to savannah and woodlands with typical AGB values of less than 85 Mg/h. In the current study, the Bouvet et al. data set was extended to regions with AGB values larger than 85 Mg/h using the methodology presented by Mermoz et al. (2014): the ESA CCI (Climate Change Initiative) land cover map was used to separate dense forest areas, over which AGB was estimated at 500 meter resolution using the results by Mermoz et al. (2015). The resulting data set will be referred to as the Bouvet-Mermoz data set in the following.

## 3   Methods

The region selected for this study was the African continent because the Bouvet-Mermoz data set, which is the only one that has been produced using SAR observations made in the same frequency band (L-band) as SMOS, is limited to Africa. The African continent contains arid, equatorial and temperate regions (Kottek et al., 2006) with deserts, shrublands, mediterranean woodlands, grasslands, savannah and rainforests (Olson et al., 2001). Therefore, this study covers a wide range of climate regions and biomes and allows to extend the analysis of L-VOD data to monitor vegetation properties, in particular biomass, at larger scales than previous studies (Grant et al., 2016; Lawrence et al., 2014; Vittucci et al., 2016).

**Unlike SMOS-IC and SMOS L3 products, which are produced natively on the 25 km EASEv2 grid, the SMOS L2 L-VOD products are provided on the ISEA4H9 grid. A spatial interpolation was required to align the SMOS L2 L-VOD to the EASE 25 grid. In order to maintain as much as possible the meaning of the opacity e.g. close to the coastline or transitions between the two grid systems, this interpolated Level 2 (hereafter iL2) L-VOD is obtained using: *(i)* whenever possible a DeLaunay triangulation linear interpolation (three valid L2 L-VOD), *(ii)* or a linear interpolation (only two valid L2 L-VOD), *(iii)* or the nearest L2 L-VOD (only one valid L2 L-VOD) to EASE25 grid point within a neighbour defined by the 25km EASE square cell.**

AGB, precipitation, tree height and MODIS NDVI/EVI data were aggregated and re-sampled to the EASEv2 grid common to the SMOS L3 and IC data sets using the Geospatial Data Abstraction Library (GDAL) routine `gdalwarp` in average mode. Regarding, the SMOS Level 2 data, several SMOS Level 2 retrievals are available for a given day for high northern and southern latitudes. At these latitudes, the best retrievals (corresponding to lower values of the cost function `Chi2`) were selected.

In spite of observing in a protected band dedicated to research observations, some radio frequency interferences (RFI) from human-built equipment affect the quality of the SMOS observations. **Several quality indicators are present in the SMOS**

L2 and L3 products. The `DQX` parameter uses the inverse linear tangent model (Jacobian) to translate the observation uncertainty (radiometric accuracy) into the parameter space uncertainty. The forward models are much more sensitive for lower values of the (SM, L-VOD) parameter space (leading to low `DQX`) than for higher values (leading to high `DQX`). Therefore, filtering to keep the lowest `DQX` implies a risk to bias our results toward lowest retrieved values, particularly

for tropical forest where both SM and L-VOD are high. In addition, the `DQX` parameter does not give information about the correctness of the solution, which is based on a quality of a fit. Therefore, the `Chi2` (goodness of the fit) was used to filter out the retrieved solutions. Several tests were done and a value of 3, corresponding approximately to the peak of the `Chi2` probability distribution was found to be a good threshold. This is in agreement with the values used in other studies (see for instance, Román-Cascón et al., 2017).

In the case of SMOS-IC, data with a root mean squared difference between modelled and observed brightness temperatures larger than 10 K were filtered out. In addition, the L-VOD time series of the three products were analysed grid point-to-grid point, and values with a deviation (in absolute value) larger than 2.5 with respect to the grid point average $\sigma$ (were $\sigma$ is the standard deviation) were considered as outliers and also filtered out.

The main evaluation strategy used in this study is to compare L-VOD data to the evaluation data sets presented in Sect. 2.

These variables such as above ground biomass, tree height, or long-term averages of mean annual precipitations are not expected to change quickly over time. The biomass data sets discussed in Sect. 2 were produced with observations done from years 1995 to 2010. The comparison of L-VOD with the other data sets was done using L-VOD **data from 2011 and 2012, as 2011 is the first complete year after the SMOS commissioning phase, which ended in June 2010. The L-VOD data for 2011 and 2012 were averaged** to avoid short-term variations due to changes in the vegetation water content over short time

periods.

To get a quantitative assessment of the correlation and the dispersion of L-VOD versus the evaluation data sets, three correlation coefficients were computed. The Pearson correlation coefficient $R$ is a measure of the linear correlation between two variables. If the relationship linking these variables is linear with no dispersion, $R$ equals 1 (both variables increase together) or -1 (one variable increases when the other decreases). However, the relationships between L-VOD and the evaluation data are

not expected to be linear in most of the cases. Therefore, the Spearman and Kendall rank correlations (which can range from -1 to 1) were also computed to quantify monotonic relationships whether linear or not (the exact definition of the Spearman and Kendall rank correlations is given in the Supplementary Information).

**The AGB and L-VOD relationship was studied for different biomes using the IGBP land cover classes (Loveland et al., 2000). Table S2 summarizes the IGBP classes and Figure S1 shows their spatial distribution using the Bouvet-**

**Mermoz AGB map. For a single biome, a linear function gives a good fit to the AGB and L-VOD relationships (see Sect. 4.3). In contrast, the global relationships linking the AGB datasets and L-VOD are significantly non-linear, therefore fits were computed following the approach used by Liu et al. (2015). The L-VOD data were binned in 0.05-width bins.**

For each L-VOD bin, the 5th and 95th percentiles and the mean of the AGB distribution were computed, providing three AGB curves as a function of L-VOD. The three curves were fitted with the function used by Liu et al. (2015),

$$AGB = a \times \frac{\arctan(b\,(vod - c)) - \arctan(-bc))}{(\arctan(\infty) - \arctan(-bc)} + d, \tag{1}$$

and with a logistic function,

$$AGB = \frac{a}{1 + e^{-b(VOD-c)}} + d. \tag{2}$$

In Eqs. 1 and 2, the parameters $a, b, c$ and $d$ are varied to get the best fit to the curves. The fitted curves give AGB in Mg/h units as a function of L-VOD, which is a dimensionless quantity. Therefore the units of $a$ and $d$ are Mg/h and $b$ and $c$ are dimensionless quantities.

## 4  Results

Figure 1 shows the average L-VOD computed over 2011 and 2012 using both ascending and descending orbits for the three SMOS L-VOD products. In addition, it also shows the standard deviation (STD) and the number of points of the local time series after applying the filters discussed in Sect 3. The three SMOS L-VOD products show a similar spatial distribution but the SMOS-IC L-VOD shows a smoother spatial distribution than the iL2 and L3 datasets. The highest values are found in equatorial forest regions and L-VOD decreases monotonically with distance to the equatorial forest in the tropical area and beyond. The STD of the L-VOD time series also increases towards the equatorial forest, in particular for the iL2 and L3 datasets. The number of points in the time series is lower for the IC dataset due to the lower revisit frequency arising from the requirement of having brightness temperature measurements spanning an incidence angle range of at least $20°$ (Fernandez-Moran et al., 2017).

Figure 2 shows the evaluation data after resampling to a 25 km EASEv2 grid: the 2011-2012 average of the MODIS NDVI and EVI indices, tree height, mean annual precipitations and AGB datasets. EVI and NDVI also decrease with increasing distance to the equator but more slowly than L-VOD. The tree height map shows two main populations: the equatorial forest, with heights larger than 20 meters, and the rest of the continent, where most of the vegetation is lower than $\sim 5$ meters. In contrast to the previous quantities, AGB can vary in two orders of magnitude, therefore AGB maps are shown in logarithmic units in Fig. 2. The Baccini, Saatchi and Bouvet-Mermoz maps show a similar AGB distribution. In contrast, the Avitabile map shows a much sharper decrease of AGB from the equatorial forest region to the rest of the continent.

### 4.1  Comparison of the three L-VOD data sets

Figure 3 shows the scatter plots of SMOS IC L-VOD with respect to the evaluation data. The scatter plots obtained with the iL2 and L3 data sets are shown in Figs. S2 and S3, respectively. A visual inspection shows that the scatter plots obtained with IC

L-VOD are significantly different than those of iL2 and L3 L-VOD, as they show smoother relationships with lower dispersion with respect to all the evaluation data sets than the equivalent plots for iL2 and L3 L-VOD.

A quantitative assessment of the correlation and the dispersion of the different scatter plots can be found in Table 1, where Pearson, Spearman and Kendall correlation coefficients are given for the three L-VOD data sets with respect to the evaluation data sets. The lowest Pearson correlation coefficient values were obtained for L3 L-VOD ($R = 0.65 - 0.87$). The Pearson correlation coefficients obtained for iL2 L-VOD are similar ($R = 0.67 - 0.87$) to those obtained for L3 L-VOD but systematically higher by up to 4%, while the values obtained for IC L-VOD are the highest ($R = 0.77 - 0.94$) with respect to all the evaluation data sets. The correlation increase is in the range of 5%-10% with respect to iL2 L-VOD and up to 15 % with respect to L3 L-VOD. **The rank correlation values with respect to all the evaluation datasets are also higher for IC L-VOD ($\rho = 0.78 - 0.91$, $\tau = 0.61 - 0.75$), followed by iL2 L-VOD ($\rho = 0.67 - 0.83$, $\tau = 0.50 - 0.65$) and L3 L-VOD ($\rho = 0.66 - 0.80$, $\tau = 0.49 - 0.62$). These results are in agreement with those obtained with the Pearson correlation and imply that the lower Pearson correlation values obtained for the L3 and iL2 datasets are not due to a correlation that could be better but more non-linear than that of the IC dataset. Therefore, using eight vegetation-related evaluation data sets and three different metrics, the most consistent SMOS L-VOD data set is SMOS-IC. This result implies that, currently, the SMOS-IC dataset is the best SMOS L-VOD product to perform vegetation studies, and the rest of the current study will focus on SMOS-IC L-VOD.**

## 4.2 Comparison of SMOS IC L-VOD to other data sets

The relationship between tree height and IC L-VOD was found to be close to linear with a high Pearson correlation coefficient ($R = 0.87$, Table 1), in agreement with previous findings using SMOS L2 data (Rahmoune et al., 2014).

With respect to visible/infra-red indices such as EVI and NDVI, Figure 3 shows that both indices saturate even for moderate L-VOD values of $\sim 0.5$, in agreement with previous studies (Lawrence et al., 2014). The correlation coefficients are $R = 0.80 - 0.81$ and $\rho = 0.86 - 0.88$ for NDVI and EVI. Regarding precipitation, the scatter plots show more dispersion ($R = 0.77$, $\rho = 0.82$) than those obtained with NDVI and EVI but there is a saturation in the mean annual precipitation values for L-VOD values higher than $\sim 0.6 - 0.7$.

Regarding the different AGB data sets, most of the scatter plots show a clear non-linear relationship between L-VOD and AGB. The relationship between Baccini et al. (2012) AGB versus IC L-VOD is the less non-linear one and the associated Pearson correlation coefficient is the highest found ($R = 0.94$, $\rho = 0.90$). The relationship between Avitabile et al. (2016) AGB and L-VOD is the most non-linear one ($R = 0.85$, $\rho = 0.84$). It shows a low sensitivity to low L-VOD values and a large dispersion for high L-VOD values with AGB ranging from $\sim$300 Mg/h to 500 Mg/h. The relationship between L-VOD and the Bouvet-Mermoz AGB data set ($R = 0.89$, $\rho = 0.91$) also shows a significant dispersion for high L-VOD values with AGB spanning a range from 200 to 400 Mg/h. In contrast, the results obtained with the Saatchi et al. (2011) ($R = 0.92$, $\rho = 0.91$) and Baccini et al. (2012) data sets show a single AGB peak for the highest SMOS L-VOD values with values of $\sim 280$ Mg/h and $\sim 320$ Mg/h, respectively. In summary, IC L-VOD shows a high sensitivity to AGB, with smooth relationships without

strong signs of saturation, in particular with respect to the AGB data sets from Saatchi et al. (2011), Baccini et al. (2012) and Bouvet-Mermoz.

To compare the relationship linking L-VOD and AGB to the relationship of other vegetation indices and AGB, scatter plots similar to those of Fig. 3 were computed using Saatchi's AGB with respect to MODIS NDVI and EVI (Fig. 4). There is a close-to-linear relationship for AGB lower than $\sim 90$ Mg/h and EVI or NDVI lower than 0.4 and 0.7, respectively. However, in contrast to L-VOD, the relationship saturates for EVI and NDVI higher than 0.5-0.6 and 0.7-0.8, respectively, for which AGB increases sharply from 90 to 300 Mg/h. This is expected as the visible/infra-red indices are sensible to the greenness of the canopy, which is not closely related to the total AGB in densely vegetated regions.

To get further insight into the global AGB versus L-VOD relationship, the fitting method described in Sect. 3 was used. Fits of the same quality were found using Liu's function (Eq. 1) and the logistic function (Eq. 2). Figure 5 shows the fits using a logistic function and Table S3 shows the best-fit parameters. Even if the overall form of the scatterplots of L-VOD and the four different AGB data sets are different, fits of the same quality were obtained for the four relationships. The Pearson correlation coefficients ($R^2$) of the fitted function with respect to the points to fit are in the range from 0.990 to 0.999 (Table S3). Eq. 2 with the best-fit coefficients of Table S3 for the "mean" curves can be used to transform SMOS IC L-VOD into AGB, while the 05th and 95th quantile best-fits can be used to provide an uncertainty interval.

## 4.3 Comparison of IC L-VOD to other data sets per land cover class

### 4.3.1 AGB data sets

Figure 6 shows the relationship of L-VOD versus the four AGB data sets (from left to right: Bouvet-Mermoz, Saatchi, Baccini; Avitabile) for different IGBP land cover classes (from top to bottom: open shrublands; croplands; grasslands; croplands and natural vegetation mosaics; savannah; woody savannah; evergreen broadleaf). Each panel of Fig. 6 shows the regression line and the corresponding equation, as well as values of the Pearson $R$, Spearman $\rho$ and Kendall $\tau$ coefficients.

Maximum L-VOD values increase from grasslands, croplands, shrublands and savannahs, where L-VOD reaches a maximum value of $\sim 0.4$, to croplands and natural vegetation mosaics and woody savannahs, where L-VOD reaches a maximum value of $\sim 0.6 - 0.7$. L-VOD values higher than $0.7$ were only found in the evergreen broadleaf equatorial forest, where the L-VOD range is $0.5 - 1.2$.

There are clear trends in the slope of the regression lines. For Bouvet-Mermoz and Saatchi AGB data sets the trends are consistent. Slopes increase from 75-86 Mg/h from shrublands and croplands to 110-150 Mg/h for grasslands, croplands and natural vegetation mosaics, savannahs and woody savannahs. Finally the AGB versus L-VOD relationship slopes increase to 215-250 Mg/h for broadleaf evergreen forest. The general trends found with the Baccini AGB data set are in overall agreement with those of Bouvet-Mermoz and Saatchi but the slopes for shrublands and grasslands are significantly lower (2-44 Mg/h) while those for croplands and natural vegetation mosaics, savannahs and woody savannahs reach 160-210 Mg/h, which are values significantly higher than the ones obtained with Bouvet-Mermoz and

Saatchi (122-156 Mg/h). The slope obtained for the evergreen broadleaf equatorial forest was in good agreement with the two other AGB datasets (265 Mg/h). On the other hand, the slopes of the Avitabile AGB and L-VOD do not show the same trends of the other three AGB data sets. Slopes for shurblands, croplands, grasslands and savannahs are so low as 13-44 Mg/h. The slope increases for mosaics of croplands and natural vegetation up to 87 Mg/h, still significantly lower than the range of 132-174 Mg/h found with the other three AGB data sets. The regression line for the scatter plot for Avitabile's woody savannah AGB increases up to 175 Mg/h, an intermediate value with respect to those found with Saatchi's (123 Mg/h) and Baccini's AGB (211 Mg/h), and actually the scatter plot shows signs of bimodality for L-VOD values of 0.5-0.7. In contrast, the slope obtained for evergreen broadleaf forest using Avitabile's AGB is much higher (362 Mg/h) than those obtained with the other three AGB data sets (215-265 Mg/h).

Many of the relationships are close to linear with Pearson coefficients $R$ up to 0.70-0.87 and similar Spearman $\rho$ values. SMOS L-VOD is well-correlated to Bouvet-Mermoz and Saatchi's AGB for all IGBP classes with Pearson correlation coefficients $R$ of $0.6 - 0.85$ (except with Saatchi AGB in shrublands, which is lower, $R = 0.49$). With respect to Baccini AGB, the Pearson correlation is high ($R = 0.7 - 0.87$) for all IGBP classes but for shrublands and grasslands, where it was found to be very low: $R = 0.03 - 0.39$. A similar behavior to that of Baccini AGB was found using Avitabile AGB, for which Pearson correlation values were also found to be low for shrublands and grasslands ($R = 0.31 - 0.44$), while they increase for savannahs and woody savannahs to $R = 0.51 - 0.56$ and to $R \sim 0.7$ for croplands, crops and natural vegetation mosaics, and evergreen broadleaf forest.

The best correlations of AGB and L-VOD were found with: *(i)* Bouvet-Mermoz AGB for Shrublands ($R = 0.64$) and Savannahs ($R = 0.81$) *(ii)* Baccini AGB for croplands ($R = 0.76$) and evergreen broadleaf equatorial forest ($R = 0.78$) *(iii)* Saatchi AGB for grasslands ($R = 0.82$). Regarding croplands and natural vegetation mosaics, the highest correlation values were obtained with Saatchi and Baccini, which gave very similar results ($R = 0.85 - 0.87$) and somewhat higher than those obtained with Bouvet-Mermoz ($R = 0.81$). Finally, for woody savannah, the highest correlation values were also obtained with Saatchi and Baccini ($R = 0.67 - 0.70$, respectively) while with Bouvet-Mermoz ($R = 0.6$) and Avitabile ($R = 0.56$) the correlation was lower. One should note that Pearson correlation values obtained with Bouvet-Mermoz for woody savannah could be degraded by the fact that for the highest values of AGB found in this class at the SMOS resolution, the AGB estimation is a mix of Bouvet and Mermoz approaches. Actually, it is noteworthy that the highest rank correlations for woody savannahs and mosaics of croplands and natural vegetation were obtained with the Bouvet-Mermoz data set ($\rho = 0.77$ and $\rho = 0.91$, respectively). In summary, except for the Avitabile AGB dataset, all the other AGB datasets performs the best, as compared to L-VOD, for a few land cover classes.

### 4.3.2 Other auxiliary data sets

Figure 7 is similar to Fig. 6 but it shows the relationship of L-VOD versus other auxiliary data sets (from left to right: tree height, NDVI, EVI and mean annual precipitations) for different IGBP land cover classes.

Regarding tree height, the slope of the regression line is 17-27 m for all IGBP classes except for shrublands, where it is 12 m. The Pearson correlation is relatively low ($\sim 0.4$) except for mosaics of croplands and natural vegetation and for evergreen broadleaf forest ($R = 0.61 - 0.73$).

Regarding the L-VOD and NDVI relationship in different biomes, the slope of the regression line increases from 0.05 in shrublands to 0.57 in grasslands and 0.87 in mosaics of croplands and natural vegetation, before decreasing again to 0.6 in savannahs, 0.36 in woody savannahs and 0.11 in evergreen broadleaf forest as NDVI saturates. It is noteworthy that no significant difference is seen on the behavior of EVI and NDVI for high L-VOD values, in spite of the "enhanced" performance of EVI with respect to NDVI pointed out by some studies (Huete et al., 2006).

Regarding the relationship of L-VOD and the average amount of annual precipitation, L-VOD increases from 0 up to $\sim 0.7$ for increasing precipitations up to $\sim 1500$ mm (values found for croplands and natural vegetation mosaics and woody savannah). In this range of L-VOD all other vegetation tracers increase as well. For instance, Bouvet-Mermoz and Saatchi's AGB increase up to 85 Mg/h and $\sim 100$ Mg/h, respectively, and NDVI and EVI increase up to $\sim 0.7$ and $\sim 0.45$, respectively (Figs. 6 and 7). The Pearson correlation $R$ and the slope of the regression line increase from 0.2-0.3 and 266-612 mm, respectively, for shrublands and grasslands to 0.4-0.65 and 1395-1914 mm for croplands, mosaics of croplands and natural vegetation and savannahs. The Pearson correlation coefficient R and the slope decrease to 0.25 and 741 mm, respectively, in woody savannahs. Finally, L-VOD values higher than 0.6-0.7, found only in evergreen broadleaf forest, are uncorrelated with the mean annual precipitation ($R = 0.04$ and slope of $-64$ mm). The mean annual precipitation could be one of the drivers of vegetation growth in drylands. In contrast, over that threshold of $\sim 1500$ mm of annual precipitations, which occur basically in the evergreen broadleaf forest, L-VOD and the other vegetation tracers are not coupled to the amount of precipitation.

## 5 Discussion

### 5.1 Sensitivity of L-VOD to AGB

As mentioned in Sect. 2, SMOS L2 and L3 products consider heterogeneous land covers inside the SMOS footprints, while SMOS-IC does not account for footprint heterogeneity. The better results obtained with the SMOS-IC data set suggests that the approach used to account for heterogeneous land covers introduce uncertainties in the Level 2 and 3 products. Nevertheless, independently of the choice of the SMOS L-VOD data set, the results showed a generally high sensitivity of L-VOD with respect to the vegetation-related variables/indices used for the evaluation, in particular with respect to AGB ($R = 0.78 - 0.94$).

The relationship between tree height and SMOS L-VOD was found to be close to linear, confirming previous findings by Rahmoune et al. (2014) using SMOS L2 L-VOD. Vittucci et al. (2016) estimated a correlation of L2 L-VOD and tree height of 0.81, which is in good agreement with the value reported here ($R = 0.79$, Table 1). However, for IC L-VOD the relationship shows even less dispersion and a higher correlation ($R = 0.87$).

The SMOS-IC L-VOD relationships with respect to NDVI and EVI were found in agreement with those discussed using SMOS L3 data by Grant et al. (2016) as there is a saturation in EVI and NDVI for high L-VOD values. In contrast, the relationships found in this study using SMOS-IC showed less dispersion than those found by Grant et al. (2016).

Regarding the comparison to AGB, Vittucci et al. (2016) discussed the relationship linking L2 L-VOD and biomass from the Carnegie Airborne Observatory (Asner et al., 2014) at 20 selected points over Peru, Columbia, and Panama spanning AGBs from ∼50 Mg/h to ∼280 Mg/h. The relationship was almost linear, in good agreement with the results discussed in Sect. 4 for SMOS IC L-VOD for evergreen broadleaf forest.

## 5.2   Comparison of L-band sensitivity to AGB to other frequencies

**This study is devoted to L-VOD as estimated from SMOS observations but it is interesting to discuss the scatter plots presented in Sect. 4.2 in comparison those obtained for other frequencies. Figure 8a shows the fits to the 5th and 95th curves obtained analysing the Saatchi AGB and L-VOD distributions (Fig. 5c). The area in between both curves was shadowed in green color. In addition, the figure also shows the fits to the 5th and 95th curves obtained analysing the MODIS NDVI and L-VOD distributions (Fig. 4). The area in between both curves was shadowed in pink color. Since the dynamic range of L-VOD and NDVI are significantly different, both quantities were normalized from 0 to 1 dividing by their maximum values (1.24 and 0.83 for L-VOD and NDVI, respectively) in order to better show the sensitivity to AGB.** As discussed in Sect. 4.2, NDVI shows some sensitivity to AGB only for low AGB values (with a low slope) before showing a strong saturation for AGB values higher than ∼ 70 Mg/h.

**Regarding the VOD estimated with higher microwave frequencies, Liu et al. (2015) discussed fits of Saatchi's AGB as a function K/X/C-VOD. They used K/X/C-VOD data in the period 1998-2002 (as mentioned in Sect. 2, the data used to compute the Saatchi et al. (2011) maps were acquired from 1995 to 2005). Liu et al. (2015) computed the 5th and 95th quantiles of the AGB distribution in VOD bins obtaining two curves giving the "envelope" of the AGB versus and VOD distribution, which is the same method that was used in the current study (Sect. 3). Figure 8b shows the fits to the 5th and 95th curves shown in Fig. S4 of Liu et al. (2015), which were reproduced using the function and the parameters given in their Eq. S2 and Table S1, respectively. The area in between both curves was shadowed in brown color. In addition, Figure 8b shows the fits to the 5th and 95th curves obtained analysing the Saatchi AGB and L-VOD distributions. The area in between both curves was shadowed in green color as in Fig. 8a.** The relationship between AGB and K/X/C-VOD shows a similar shape to that of AGB versus L-VOD but it is somewhat shifted to higher VOD values. AGB increases from ∼ 50 Mg/h to ∼ 300 Mg/h for K/X/C-VOD values higher than ∼ 0.7. In contrast, the relationship between AGB and L-VOD shows a more steady increase from low to high AGB and L-VOD values. In particular, it does not show a threshold beyond which the relationship saturates and the slope increases significantly. **One must bear in mind that the time periods of the data used to compare with K/X/C-VOD are not the same, as the L-VOD period used in this study is 2011-2012 and that more detailed comparisons of the sensitivity to AGB of VOD at different frequencies would be interesting. However, the non-linearity of the curve and the difference sensitivity to high AGB from different frequencies is driven by the high AGB values in the dense equatorial forest, which is not supposed to vary strongly in a few years time at the**

SMOS spatial resolution. In addition, it is worth noting that the different shapes of the L-VOD and AGB relationships with respect to the K/X/C-VOD and AGB relationships are in agreement with what it is expected from the radiation transfer theory (Wigneron et al., 1995, 2004; Ferrazzoli and Guerriero, 1996) and previous results on L-VOD and X/C-VOD comparison by Grant et al. (2016) and Vittucci et al. (2016) as Fig. 8b shows that for a given AGB, L-VOD is lower than VOD at higher frequencies, as expected.

## 5.3 Comparison of the different AGB datasets

Estimating the AGB from remote sensing measurements is complex and the errors of different retrieval methods are not easy to characterize. Interpreting why L-VOD compares better to a given AGB data set for a given IGBP class (Sect. 4.3) is not easy.

The Avitabile AGB data set shows a sharp decrease from the Equatorial region with distance is not seen in any other AGB map nor in the L-VOD maps. Avitabile AGB and L-VOD scatter plots are also significantly different to those computed with the original Baccini and Saatchi maps. For instance, the low AGB versus L-VOD slopes obtained for low shrublands, grasslands and croplands are much lower than those found with the original Saatchi and Baccini datasets. The scatter plot found with Avitabile for woody-savannah resembles an overlay of the scatter plot obtained with Baccini and the scatter plot obtained with Saatchi. Finally, the slope of the AGB versus L-VOD in evergreen broadleaf forest is $\sim 30\%$ higher than those found with the other data sets. The singular behavior of Avitabile AGB could arise from the fact that it is a pure data driven method and that it is therefore very sensitive to the data used to train the method. In the Avitabile et al. (2016) training database, high AGB plots could be over-represented.

On the other hand, as mentioned in Sect. 4.3, the distribution of Baccini AGB for woody savannah is significantly different to the other datasets, which much higher values than those found for Bouvet-Mermoz and Saatchi AGB. Actually, with Baccini AGB, the value of the slope obtained for woody savannah is 80 % of that obtained for evergreen broadleaf forest, while this ratio is only $55\%$ for Bouvet-Mermoz and Saatchi AGB. This high slope for woody savannah is responsible of the lower non-linearity of the global AGB and L-VOD relationship using the Baccini data set. Woody savannah in the IGBP classification is defined as herbaceous vegetation and a forest canopy cover between 30 % and 60 %. AGB could be overestimated in this heterogeneous land cover class in the Baccini dataset due to the fact that no microwave data but only MODIS is used for the spatial extrapolation (Sect. 2). Figure S4 shows scatter plots of the four AGB datasets as a function of the Simard et al. (2011) tree height estimation. The relationship is almost linear for Baccini et al. (2012) AGB, which is not the expected behavior from allometric relations (Chave et al., 2014).

Radar observations in low vegetation regions such as shrublands and grasslands are thought to be very sensitive to biomass variations, in spite of a significant sensitivity to soil moisture. The high correlation of the two AGB maps involving radar data, either as the main source of information (Bouvet-Mermoz) or together with optical and elevation data (Saatchi), with SMOS L-VOD in grasslands would confirm this fact, as the high correlation in shrublands for Bouvet-Mermoz. The low slopes found for shrublands and grasslands when comparing to Baccini AGB also support this interpretation. Interestingly, the Bouvet-Mermoz AGB data set, which has been obtained from L-band SAR data

and is the only one developed with a particular focus on savannahs, shows a linear relationship between L-VOD and AGB with a very low dispersion.

## 6 Conclusions

Three different SMOS-based L-VOD data sets were evaluated and compared to precipitation, tree height, NDVI, EVI and AGB data. Lower dispersion and smoother relationships were obtained by using SMOS-IC L-VOD, compared to the iL2 and L3 L-VOD data sets. Consistently, the rank correlation values obtained with SMOS-IC were significantly higher by 5-15 % than those obtained with Level 2 and Level 3 L-VOD data sets.

The relationships between AGB estimates and L-VOD were strong ($R = 0.85 - 0.94$) but differed among the products. For low vegetation classes (grasslands to woody savannah), the best performance was achieved with the Bouvet-Mermoz, **Baccini and Saatchi biomass data sets. The biomass data by Baccini and Saatchi showed the best agreement with L-VOD for dense forest ($R = 0.70 - 0.79$). Avitabile's AGB data showed low correlation values with L-VOD for low vegetation classes and a similar performance to Bouvet-Mermoz for dense forest ($R = 0.64 - 0.67$). The AGB and L-VOD relationships can be fitted over the entire range of both variables with a single law using a sigmoid logistic function. However, an analysis per land cover class showed that within the same land cover class, the L-VOD and AGB relationship is close to linear.** Therefore, the global non-linear relationship, found when all the different land cover are considered together, arises from different slopes in the L-VOD/AGB relationship obtained for different land cover classes considered separately. For low vegetation classes, the annual mean of L-VOD spans a range from 0 to 0.7 and could be related to the mean annual precipitations.

The relationship between AGB versus L-VOD was compared to the ones between AGB versus NDVI and AGB versus K/X/C-VOD from Liu et al. (2015). As expected, NDVI saturates strongly and it becomes weakly sensitive to AGB changes from $\sim 70$ to $\sim 300$ Mg/h. With respect to K/X/C-VOD, the AGB also increases slowly as VOD increases for most ($\sim 70$ %) of the K/X/C-VOD dynamic range but **it saturates for VOD** $> 0.8$. In contrast, AGB values show a steady increment as L-VOD increases over the whole L-VOD dynamic range.

The equations computed in this study can be used to estimate AGB from SMOS-IC L-VOD. Of course, these equations depend on the data set used as reference to fit the AGB and L-VOD relationship. Three of them (those determined with Baccini et al. (2012), Saatchi et al. (2011) and Bouvet-Mermoz) gave very similar values when the 5th and 95th percentiles of the distributions were taken into account.

The results obtained in this study showed that the L-VOD parameter estimated from the SMOS passive microwave observations is an interesting index to monitor AGB at coarse resolution ($\sim 40$ km). **Despite its coarse spatial resolution, the advantage of using SMOS L-VOD is that it is possible to compute one AGB map per year, for instance, which allows to perform temporal estimations of the changes in the global carbon stocks at large scales (Brandt et al., 2018).**

**Table 1.** Pearson's $R$, Spearman's $\rho$ and Kendal's $\tau$ correlation coefficients of the three SMOS L-VOD data sets with respect to mean annual precipitations, tree height, MODIS NDVI and EVI and AGB from Saatchi et al. (2011), Avitabile et al. (2016), Baccini et al. (2012) and Bouvet-Mermoz.

| | $R$ | | | $\rho$ | | | $\tau$ | | |
|---|---|---|---|---|---|---|---|---|---|
| | IC | iL2 | L3 | IC | iL2 | L3 | IC | iL2 | L3 |
| Precipitations | 0.77 | 0.67 | 0.65 | 0.82 | 0.72 | 0.69 | 0.62 | 0.53 | 0.50 |
| Tree Height | 0.87 | 0.79 | 0.78 | 0.78 | 0.67 | 0.66 | 0.61 | 0.50 | 0.49 |
| NDVI | 0.81 | 0.75 | 0.73 | 0.88 | 0.81 | 0.78 | 0.72 | 0.63 | 0.60 |
| EVI | 0.80 | 0.74 | 0.73 | 0.86 | 0.79 | 0.76 | 0.69 | 0.60 | 0.57 |
| Avitabile | 0.85 | 0.78 | 0.78 | 0.84 | 0.73 | 0.72 | 0.65 | 0.54 | 0.53 |
| Baccini | 0.94 | 0.87 | 0.87 | 0.90 | 0.80 | 0.77 | 0.74 | 0.62 | 0.60 |
| Saatchi | 0.92 | 0.84 | 0.84 | 0.91 | 0.82 | 0.80 | 0.75 | 0.64 | 0.62 |
| Bouvet-Mermoz | 0.89 | 0.81 | 0.81 | 0.91 | 0.83 | 0.80 | 0.75 | 0.65 | 0.62 |

*Author contributions.* NJRF, AM, YK and JPW planned the research discussed in this manuscript and NJRF and AM performed most the computations. SM, AB and TLT provided the AGB data sets and expertise on AGB estimations. AM, JPW and AAY provided the SMOS-IC data. PR preprocessed the SMOS Level 2 data. TK, AAB and MB reviewed the system design and the results, in particular regarding the analysis per land cover classes. All authors participated in the writing and provided comments and suggestions.

5  *Competing interests.* The authors declare that they have no conflict of interest.

*Acknowledgements.* N.J.R.-F, J.-P.W, T.K., A.M. and Y.H.K. acknowledge partial support by the ESA contract No. 4000117645/16/NL/SW Support To Science Element SMOS+Vegetation and by CNES (Centre National d'Etudes Spatiales) TOSCA program. They also acknowledge and interesting discussions with other colleagues involved in the SMOS+Vegetation project (Marko Scholze, Matthias Drusch, Michael Vossbeck, Wolfgang Knorr, Cristina Vittucci and Paolo Ferrazzoli). SMOS Level 3 data were obtained from the "Centre Aval de Traitement 10  des Données SMOS" (CATDS), operated for the CNES (France) by IFREMER (Brest, France).

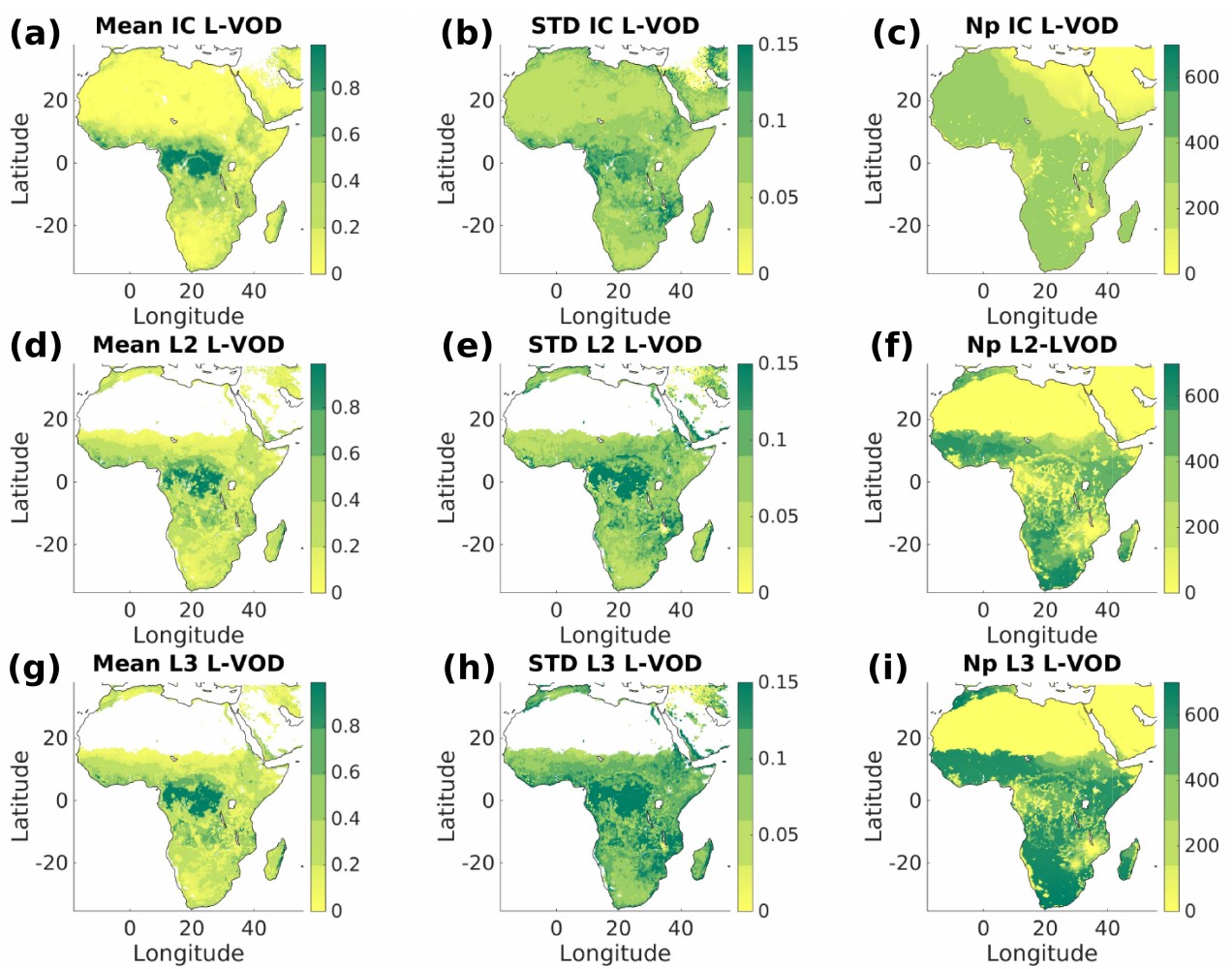

**Figure 1.** Years 2011-2012 average of L-VOD for the SMOS-IC, SMOS iL2 and SMOS L3 data sets (panels **a**, **d** and **g**, respectively), corresponding standard deviation (STD, panels **b**, **e** and **h**) and number of points ($N_p$, panels **c**, **f** and **i**) after filtering (Sect. 3) of the local L-VOD time series for the three products.

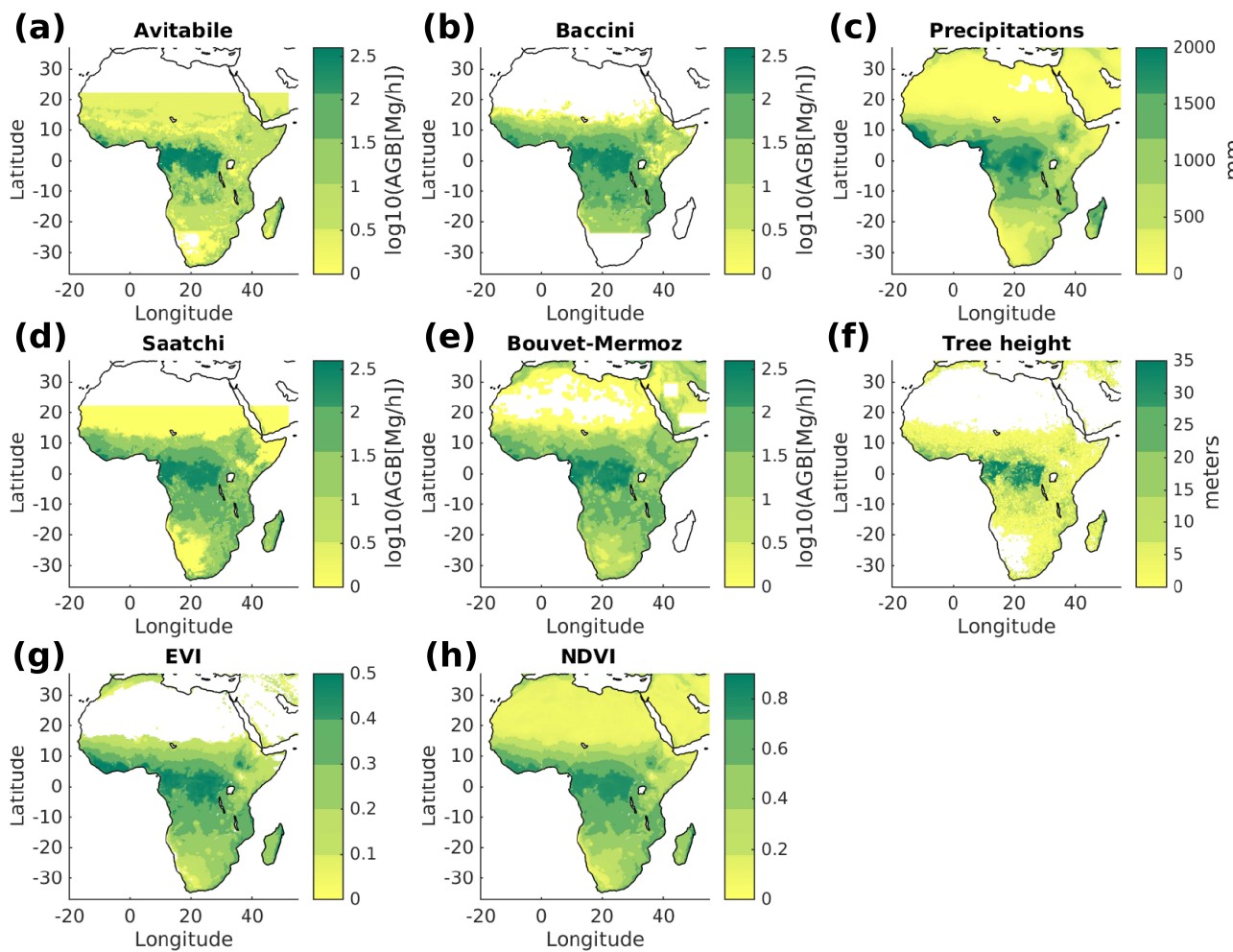

**Figure 2.** AGB maps from Avitabile et al. (2016), Baccini et al. (2012), Saatchi et al. (2011) and Bouvet et al. (2018) (panels **a, b, d, e**, respectively). Mean annual precipitations and tree height (panels **c, f**, respectively). Year 2011-2012 average of MODIS EVI (**g**) and NDVI (**h**).

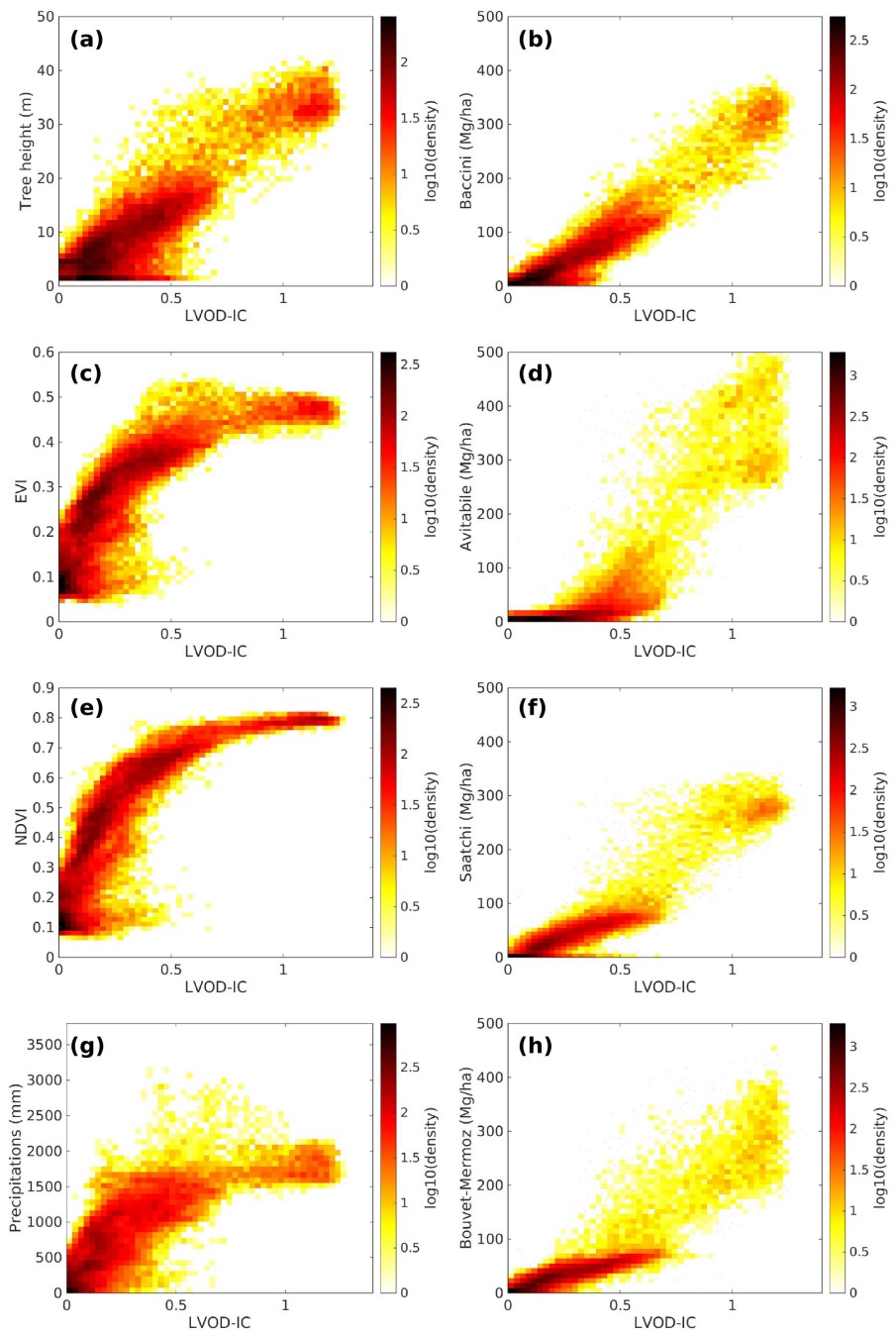

**Figure 3.** Density scatter plots of SMOS-IC L-VOD respect to: tree height (**a**), EVI (**c**), NDVI (**e**), cumulated precipitation (**g**), Baccini et al. (2012) AGB (**b**), Avitabile et al. (2016) AGB (**d**), Saatchi et al. (2011) AGB (**f**) and Bouvet-Mermoz AGB datasets (**h**).

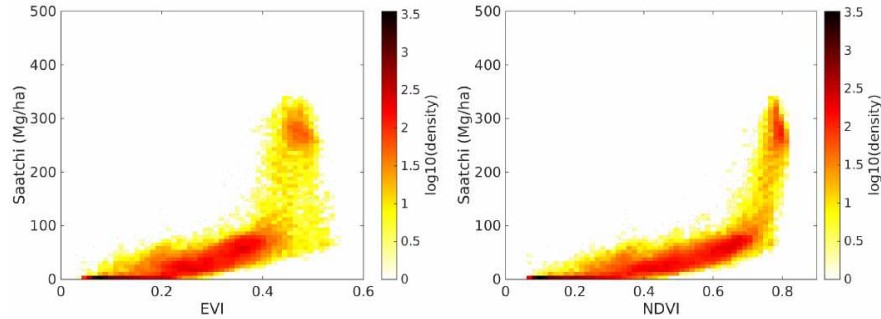

**Figure 4.** Scatter plots of MODIS NDVI and EVI with respect to Saatchi et al. (2011) AGB.

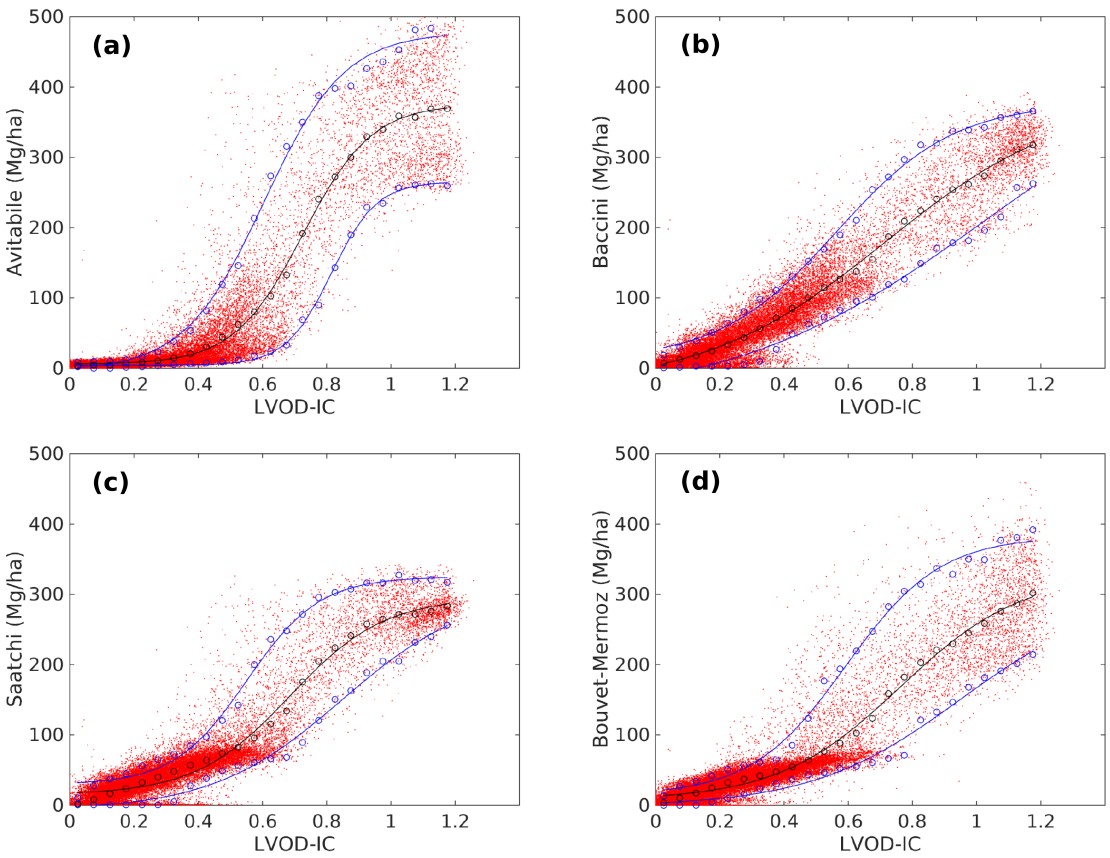

**Figure 5.** AGB versus L-VOD scatter plots of Fig. 3 but plotted as point scatter plots. In addition, on the right-hand panels, the 5th and 95th percentiles of the AGB distribution in bins of L-VOD are displayed as blue circles while the mean is displayed as black circles. Solid blue and black lines are the fits obtained using a logistic function (Eq. 2) with the parameters given in Table S3 for the 5th and 95th percentiles and the mean curves.

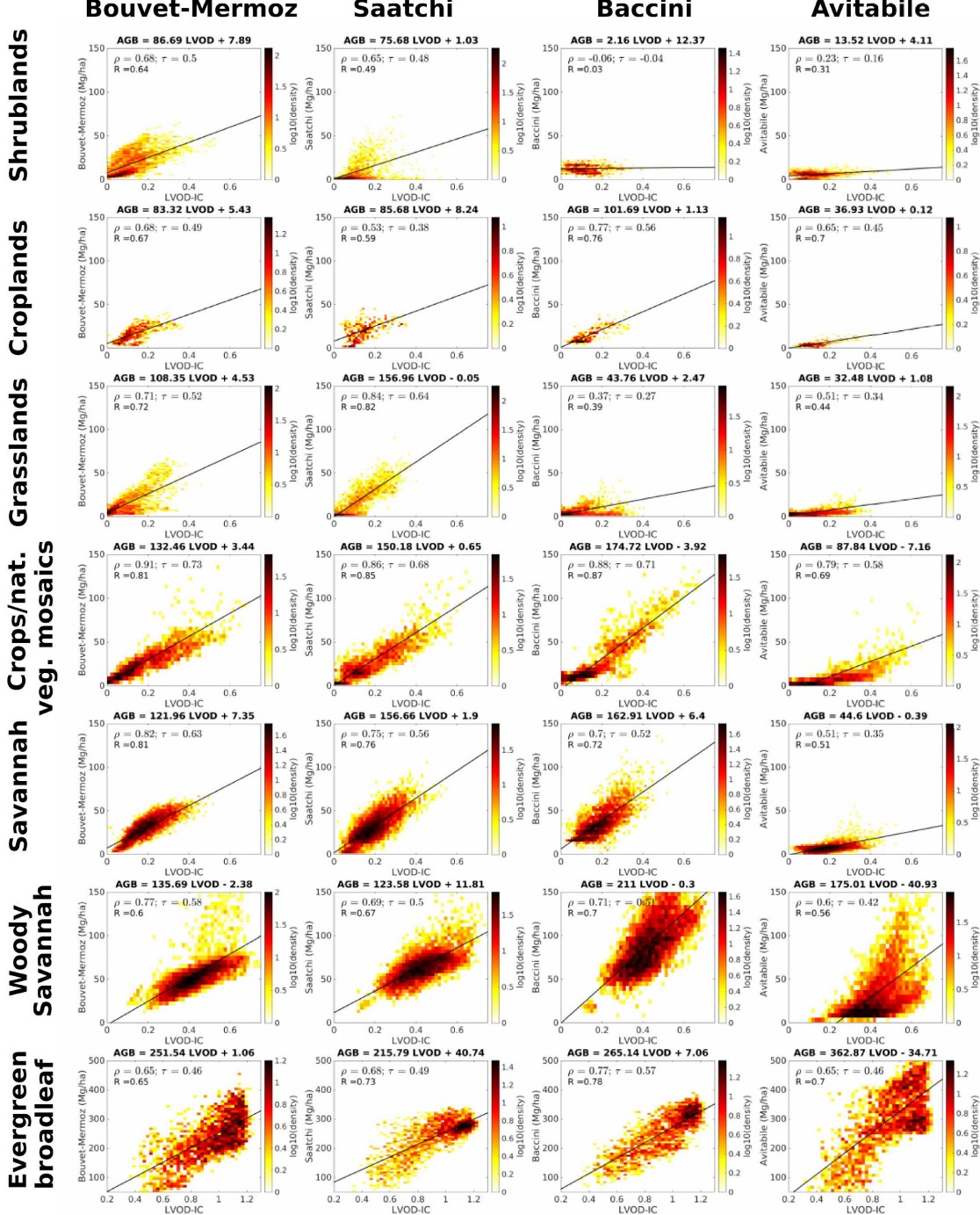

**Figure 6.** SMOS IC L-VOD relationships versus the four AGB data sets (from left to right: Bouvet-Mermoz, Saatchi, Baccini; Avitabile) for different IGBP land cover classes (from top to bottom: open shrublands; croplands; grasslands; croplands and natural vegetation mosaics; savannah; woody savannah; evergreen broadleaf). Each panel shows the regression line and equation, and values of the Pearson $R$, Spearman $\rho$ and Kendall $\tau$ coefficients.

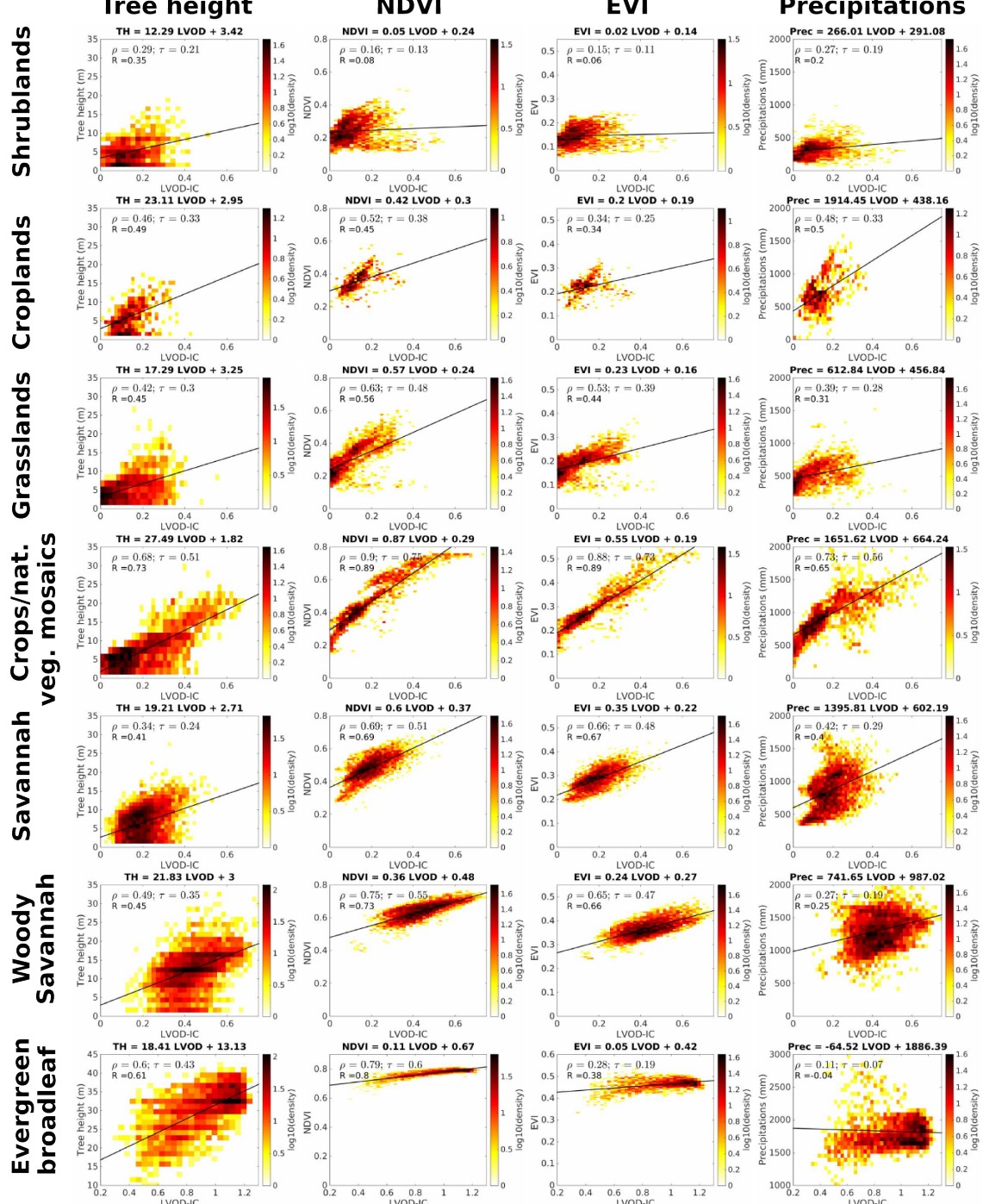

**Figure 7.** SMOS IC L-VOD relationships versus auxiliary data sets (from left to right: tree height, NDVI, EVI and average annual precipitations) for different IGBP land cover classes (from top to bottom: open shrublands; croplands; grasslands; croplands and natural vegetation mosaics; savannah; woody savannah; evergreen broadleaf). Each panel shows the regression line and equation, and values of the Pearson $R$, Spearman $\rho$ and Kendall $\tau$ coefficients.

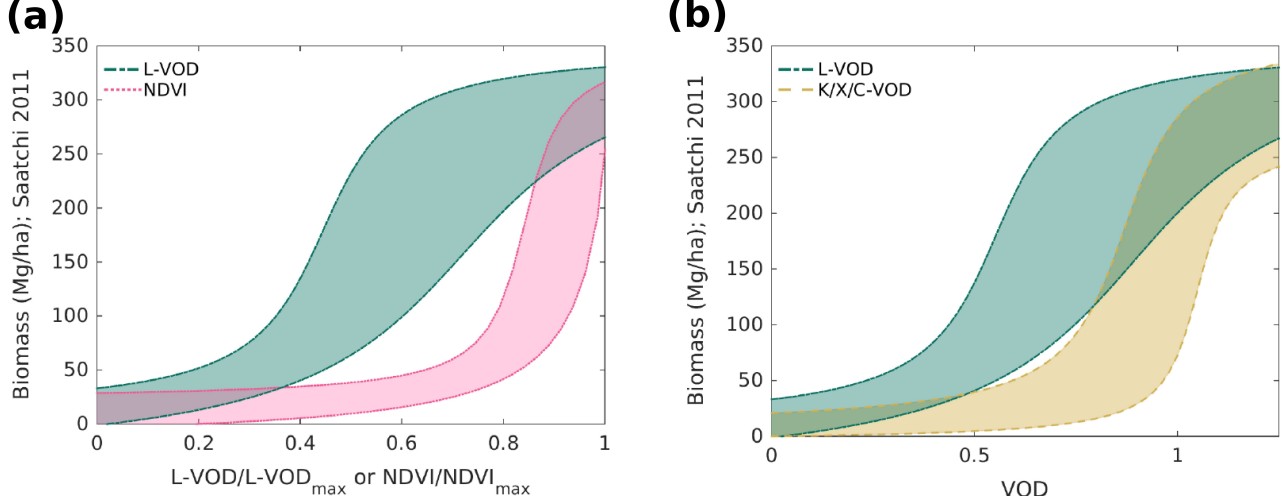

**Figure 8.** Left: Fits of the 5th and 95th percentile curves of the Saatchi et al. (2011) AGB with respect to SMOS-IC L-VOD (green) and NDVI (pink). To plot both distributions with the same scale, VOD and NDVI were normalized from 0 to 1 using their respective maxima (0.83 for NDVI and 1.24 for L-VOD). Right: Fits of the 5th and 95th percentile curves of the Saatchi et al. (2011) AGB with respect to SMOS-IC L-VOD (green) overlaid in the K/X/C-VOD versus Saatchi et al. (2011) AGB curves of Fig. S4 from Liu et al. (2015) (brown). No normalization is needed in this case as both VODs span a similar range of values.

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
