# Peer review of "An evaluation of SMOS L-Band vegetation optical depth (L-VOD) data sets: a high sensitivity of L-VOD to above-ground biomass in Africa"

_Biogeosciences, 2018_

## Referee Comment (RC1) · Anonymous Referee #1 · 26 Mar 2018

The authors present an interesting work addressing the sensitivity of SMOS L-band vegetation optical depth (VOD) to biomass. The study is centered in the African continent and employs the three available data sets of SMOS VOD data currently available (L2,L3 and IC). As independent data sets, the authors chose four above-ground biomass sources (AGB), lidar-based tree height, MODIS vegetation indices and cumulated precipitation. The differences of the three SMOS products are clearly detailed and discussed. The analyses relating VOD to the independent data sets are performed in a scientifically sound manner. However, sometimes the pretensions of the authors with respect to the obtained results are too high, specially taking into account that they are only using one year of SMOS observations. Also, they report a higher sensitivity

to AGB at L-band than at higher frequencies (K/X/C-bands), but they do not present a clear comparison of the different data sets. Therefore, I recommend this manuscript for publication after addressing the following issues:

1. The results should be re-organized in a more clear and structured way to facilitate readability and comprehension. There are many general references to relevant results in supplementary material that should directly point to a specific figure or table and be commented in the main text. Some choices made in the analysis and presentation of results are unclear (e.g. the stratification per land cover in two biomes should be further justified) and it is hard to follow the results presented in the main and the supp. material.

2. It is unclear how the authors obtain the results plotted in Fig. 4. It seems they do not use K/X/C-VOD data from year 2011 for a fair comparison to the presented results with L-band and NDVI. Instead, they show the results from Liu et al 2015, which are based on VOD time series from 1993 to 2012 and a significantly different approach. I believe the data is not directly comparable and the result presented in the figure is therefore misleading. I strongly suggest the authors to either a) include the K/X/C-VOD data from the same study period (yearly average) and detail in the methods or b) focus on the comparison of L-band VOD and NDVI and AGB. I would particularly encourage the latter. Also, the results on Fig 4 could be shown for the four different AGB data sets used in the study, for completeness.

3. The title is too ambitious and general. The focus is clearly on SMOS L-band VOD and biomass, but the results presented (using 1 year of observations over Africa) do not support the use of the words "high sensitivity". I would recommend the authors to provide a more specific title, more representative of its contents.

4. Section 5 "Discussion" is too short. Results are already discussed in Section 4, and Section 5 adds a brief overview and a comparison to literature studies. I would recommend the authors to re-organize the manuscript and include the content of Section 5

either in the results or in the conclusions as "Discussion and Conclusion".

Here is a list of more specific comments and recommendations:

1. Abstract, last sentence. Consider changing "index" by "indicator"

2. Page 2, lines 9-11. In presence of vegetation, part of the soil emission is absorbed and scattered. There are two microwave vegetation parameters that are used in the physical model to account for the effect of vegetation: the vegetation optical depth and the single effective scattering albedo. The authors should introduce here the albedo parameter, or at least mention it.

3. Page 2, line 16. Specify how "thick" is the vegetation layer that microwaves penetrate, and introduce here a comparison between frequencies (this is later briefly discussed in line 30). Is the soil emission from tropical and boreal forests reaching the satellites operating at C/X/L bands? Add references and a brief discussion to support and clarify how the different frequencies are complementary.

4. Page 3, first paragraph. Literature on SMAP L-band VOD is totally missing and should be added. For instance, a global comparison of SMAP VOD to lidar-based vegetation height is reported in Konings et al. 2017. A.G. Konings, M. Piles, N. Das, D. Entekhabi, L-Band Vegetation Optical Depth and Effective Scattering Albedo Estimation from SMAP, Remote Sensing of the Environment, Vol. 198, pp 460-470, 2017.

5. Page 3, line 17. It should be relevant to (at least) mention briefly the difference between active and passive microwave sensing of vegetation.

6. Page 3, line 25. Please, add a reference to support that the quality of the ascending data is better than the descending. I would "a priori" recommend to use both to increase coverage.

7. Page 4, line 7. SMOS is first introduced as a full-polarization radiometer but here it is stated that only dual-polarization measurements are used in the retrievals. Why? Too much information to constrain retrievals? Consider including a reference here.

8. Page 4, line 14. The authors mention that previous L-VOD retrievals are used to constrain new retrievals. How many closest retrievals? Please, be more specific.

9. Table S1. It would be relevant to include how albedo and soil roughness are computed in the different products. Also, please detail previous retrievals. ISEA should be ISEA4h9.

10. Page 5, line 6. Mention how SMOS-IC is initialized and refer to Table S1.

11. Page 5, line17. A reference is needed for Worldclim data.

12. Page 5, line 21. change "sential" by "essential" (?)

13. Page 5, line 24. Consider adding a refernce for EVI and its main differences to NDVI.

14. Page 5, line 7. Words "In a second step" are used in lines 5 and 7.

15. Page7, lines 15-16. It seems here that the authors hypothesize the AGB data set derived from L-band SAR is probably the more appropriate and therefore they restrict the study to its coverage (i.e. Africa). However, best results are obtained with Saatchi. The authors should better elaborate on why it is important to use this data set and reformulate this sentence.

16. Page 7, line 24. Please, specify which parameter is used to select the lower values of the cost function (chi-square?)

17. Page 7, line 28. There are different criteria to filter out the quality of SMOS observations. As a common practice, the DQX parameter is used. However, the authors here propose to use the Chi2 parameter larger than 3. A reference should be added to support this criteria.

18. Page 7, line 30. It would be important to show a map with the final number of samples used per pixel, after the filtering criteria is applied. It would also be relevant to show a map of the standard deviation of the estimates (apart from the average on Fig.

1). This is critical, since the study is based on a final comparison of spatial maps.

19. Page 8, line 3. The authors average on a yearly basis since they chose only one year of observations. A seasonal study would be interesting, but of coarse more years would be needed. The choice of using only one year of SMOS observations should be further justified. Also, the impact of using one year in the results should be (at least) discussed later in the manuscript.

20. Page 8, line 13. Please be specific as to the contents you are referring to in the supplementary information.

21. Page 4, line 15. It would be relevant to detail the function used for the fitting in the main manuscript.

22. Page 8, line 22. Please, detail "the remaining static data sets" and comment on Figure 1 (e.g. main visual differences between the VOD products and the AGB ones)

23. Figure 1. The reference to Mermoz is missing.

24. Page 9, line 1. Comment on Spearman and Kendall results, which confirm the results obtained with Pearson.

25. Page 9, line 20. It is interesting that only with Saatchi and Baccini there is a single AGB peak corresponding to the higher VOD values. Why do the authors believe this peak is not appearing as clearly with the other two data sets? Is it consistent that the peak is higher for Baccini than for Saatchi? The authors should elaborate on the results presented.

26. Page 9, line 22. It seems to me that also Saatchi shows a very low dispersion for low AGB values, but the plot is too small. Please, address.

27. Page 9, line 29. The authors aggregate the data sets in two groups of biomes. This separation should be further justified. Also, there are many results shown in the supplementary material that are relevant and should at least be discussed in the text.

28. Section 4.9. I would suggest the authors to include a box plot with the SMOS IC VOD results per land cover. It will give a general idea of the dispersion and the mean values of VOD per land cover. Perhaps it would also be good to show the box plots for the AGB data sets.

29. Section 5. It would be nice to add a discussion on the consistency of the four AGB data sets and on why best correspondence is found between L-VOD and the approach of Saatchi (and not the one of L-band SAR).

30. Figure 3. It would be interesting to know the number of pixels in the two groups of biomes, and whether they are balanced. Are all the correlation significant? To what level? This is important information that should be included either in the figure or the text.

31. Page 10, line 3. The authors should comment on the slope of NDVI per land cover and most relevant aspects shown in the supplementary information.

32. Page 10, line 17. Please, specify which part of the supplemtary information is being referred to here.

33. Figure 4. Legend reads "C/X VOD" but caption reads "K/X/C VOD". Please, correct.

---

## Referee Comment (RC2) · Anonymous Referee #2 · 28 Mar 2018

The study is aimed to introduce the sensitivity of the vegetation optical depth (VOD) at L band to the biomass. Different SMOS datasets, produced by different algorithms, are compared to some above ground biomass (AGB) datasets over Africa. The analysis is carried out to show the higher correlation of the L band VOD with respect to higher frequencies VOD and optical vegetation indices. The paper also presents the correlation of the SMOS VOD with other parameters like tree height and cumulated precipitations.

General comments:

The study's goal is well defined in the paper introduction where the authors claim that the retrieval of the VOD at L band can provide an important tool for the monitoring of

the vegetation properties at large scales. In the first section of the manuscript is highlighted that, besides optical measurements, passive microwave observations acquired by the SMOS radiometer can provide an important complementary information to infer the state of vegetation. Here, several references are correctly reported to introduce the study and it is emphasized how the L band observations are less attenuated through the vegetation canopy. Therefore, L band VOD is expected to sample the vegetation layer up to higher biomass values compared to higher frequency observations. This aspect represents the key point of the manuscript and it is supported by the figure 4 of the results section. Anyway, just few comments are deserved to this point while a deeper explanation of the high sensitivity of the L band should be provided in the last section of the results. Moreover, it seems that the presented research is a progress of a previous work in which some of the authors have already addressed the topic in 2016, including some results about the SMOS VOD sensitivity to tree height and AGB. I would suggest citing also this preliminary study in the introduction (doi 10.1109 / IGARSS.2016.7730383). Another general concern it is related to the use of three different VOD datasets derived from the SMOS data (L2, L3 and SMOS IC) that could confuse the reader. In my opinion, this point of view is interesting but can defocus the attention from the study objective, that it is claimed in the manuscript title. In some parts of the article it seems that too much importance is given to the intercomparison of the different VOD retrieval algorithms, instead of supporting the relevance of the VOD at L band for AGB monitoring. Furthermore, a potential user of SMOS data, could ask himself what is the product to adopt between the L2, L3 and SMOS IC for vegetation monitoring, since the strengths and weaknesses of the different approaches can be highlighted more clearly. A suggestion to address this point could be to provide a general overview of the specific aims of the different products and maybe to update the title of the research to highlight that different L-band products are compared. Despite these general issues I believe that the topic is relevant, the results are obtained with a sounding scientific approach and the supporting figures and tables are clear. Therefore, I would recommend the paper for publication after a careful revision process.

[Figure]

Specific comments:

In the section 2, "Data", is introduced the SMOS mission and the three different algorithms, considered to retrieve the L band VOD from the SMOS brightness temperature. At line 28 of page 2 is stated that only ascending orbits are considered in the study but the declaimed better overall quality of ascending pass acquisitions appears not justified. Therefore, the authors should provide a better explanation about this important constrain. In the following subsection are introduced the ESA L2 algorithm, the CATDS L3 algorithm and the INRA-CESBIO algorithm that were applied to obtain three different L band VOD data sets. If the Authors are inclined to stress the intercomparison between the outcomes of the different retrieval approach, a deeper discussion about the different algorithm could be effective to introduce the subsequent results, i.e. figure 1 and table 1. This choice, could be a good solution to solve some ambiguities between the study aim, as claimed on the paper title, and the interesting overview of the different algorithms performances. Anyway, a better explanation on the assumptions (i.e. soil roughness and albedo) under which the three different algorithms are based should be provided. After the introduction of the VOD datasets the different benchmark sets are presented. In the section 2.2.1 it is introduced the Worldclim data set, that is used to infer the relationship between the L band VOD and the mean annual precipitation. This analysis seems meaningless since, as it is reported at line 15 of page 5, the considered precipitation is extracted from a dataset ranging only between "1970-2000". This point should be clarified also considering that the relationship between the precipitation and the VOD are not well commented in the paper. In the section 2.2.4 are presented the different AGB datasets considered as benchmarks. Here the sentence "This study used four static AGB benchmark maps (Baccini et al., 2012; Saatchi et al., 2011; Avitabile et al., 2016; Bouvet et al., 2018) each with specific strengths and limitations to assess L-VOD's ability to reflect aboveground biomass in different" is questionable and not well supported by the results. In particular, the Avitabile dataset is obtained by the fusion of the Baccini and Saatchi maps through a machine learning approach and it is proved to outperform the previous datasets in terms of retrieved AGB

accuracy. The Authors should argue better the aspects related to the analysis carried out with these three different AGB data sets. On the contrary the consideration of the Bouvet dataset is very interesting and should be emphasized. In the Results section it should be provided a deeper explanation of the research outcomes, in particular the scatterplots reported in figure 2 need to be better commented.

---

## Referee Comment (RC3) · Anonymous Referee #3 · 7 Apr 2018

The paper provides evidence that the vegetation optical depth VOD derived from passive microwave satellite data at L-band frequency has strong correlation with the aboveground biomass and can be used to monitor vegetation status. The paper is well-written and the methodology and results are sound and at the same intriguing, suggesting VOD as a potential satellite derived parameter to explore in future studies. I recommend the paper for publication but I have few suggestions and recommendations that may help improve the interpretation of the results before final publication of the paper.

1. The paper does not provide a strong motivation of what VOD can be used for. Veg-

etation aboveground biomass is one of the most important global ecosystem variable for carbon cycle and climate mitigation. However, the strong correlation of VOD with biomass does not necessarily mean VOD from passive microwave at approximately 0.5-degree resolution is useful for biomass estimation or monitoring. VOD can be used to monitor vegetation water content at regional scales given its coarse resolution and frequent observation. I would like to suggest that although the authors correlate the result with biomass, they emphasize the use of VOD for monitoring vegetation water content. Biomass and water content are similar in magnitude with biomass being more static and water content more dynamic. 2. The method says: "The main evaluation strategy used in this study is to spatially compare L-VOD to the evaluation data set." Although the pixel values are extracted from all the data sets to compare. However, this is not a spatial analysis because the spatial information almost disappears in the correlation studies. Unless a specific spatial correlation model was used to capture the pattern. Some of the vegetation classes are separated that can help with spatial variation of the data sets but again this is only a simple correlation study and does not include spatial analysis of data sets. 3. Figure 2. The density scatter plots with multiple parameters show that there is a strong relationship between VOD and all the parameters. Some of the most interesting ones are the optical data and precipitation showing a strong saturation with respect to VOD suggesting that VOD can be used as a complementary measurement to look at the vegetation. Wavelength is probably the most powerful aspect of the VOD measurements compared to optical data. If VOD correlated with EVI and NDVI over the entire range, then the interpretation of VOD could've been more difficult. I recommend the authors discuss this in the paper. 4. The relationship between VOD and biomass from different products are interesting. The fact that L-band VOD does not show a clear saturation with biomass may be due to: a. 1. At very coarse resolution (40-50 km), the variations of forest biomass on the landscape is dominated with the landscape heterogeneity. Larger heterogeneity (e.g. forest/non-forest mixture) will improve the relationship of VOD with biomass. This may mean that the VOD is also co-varying with the vegetation cover. In fact, the straighter

relationship with Baccini data is the artifact of this effect. Baccini biomass is strongly correlated with MODIS VCF (vegetation continuous field) data and therefore causes a more linear relationship. Whereas other maps and including the vegetation height from Simard do not show this linear relationship. There is no reason for VOD and biomass to have a linear relationship. I recommend the authors discuss this point and may even include the MODIS VCF product as a layer similar to NDVI in the mix. b. At coarse resolution, the global biomass values are much smaller on the average. Biomass at 1-ha can reach a very large number at some ecosystems. However, at 40 km as it is mixed with the heterogeneity the average is almost smaller. This is one more reason for better sensitivity to biomass. However, it would be interesting to focus on different range of biomass with VOD. c. Over Africa, all dense tropical forests are clustered around 300 Mg/ha of biomass on the graphs in figure 2. If the goal of the paper is sensitivity to biomass, it may not be a bad idea to separate areas of up to 150 Mg/ha that includes the first cluster from the second cluster and study it separately. The binary feature of biomass in Africa, from woodlands to dense humid tropical forests in area may introduce a false strong correlation with biomass that need to be discovered further. Figure 3 is supposed to show this effect. However, the authors mix this up with precipitation and NDVI and only show the result from Bouvet. It would be good to show this for all biomass maps so the variations of the relationships are discussed. d. Although the paper is written for the biogeoscience community, it would be important for the authors to provide some explanation of why L-band data from passive measurements may have better relations with biomass compared to active measurements at the same frequency. e. How different are the relationships between VOD and different biomass maps and how can the difference be interpreted? f. In table 1, there are three metrics to show the relations between VOD and biomass and other parameters. However, only Baccini result is highlighted in the abstract. Why? The table does not necessarily support this. Furthermore, there is not physical reason that the scattering or emissivity has to be linearly related to biomass. 5. Figure 4 is a bit difficult to understand. The colors and what the legend provide cannot be easily deciphered. It seems

one should the see the saturation of NDVI and a much linear relationship with VOD but I am not sure the figure explicitly shows this. I recommend either making the figure a bit simple or provide more information in the caption and change colors so the points are clear.

---

## Author Comment (AC1) · 3 May 2018

**Referee # 1**

The authors present an interesting work addressing the sensitivity of SMOS L-band vegetation optical depth (VOD) to biomass. The study is centered in the African continent and employs the three available data sets of SMOS VOD data currently available (L2,L3 and IC). As independent data sets, the authors chose four above-ground biomass sources (AGB), lidar-based tree height, MODIS vegetation indices and cumulated precipitation. The differences of the three SMOS products are clearly detailed and discussed. The analyses relating VOD to the independent data sets are performed in a scientifically sound manner. However, sometimes the pretensions of the authors with respect to the obtained results are too high, specially taking into account that they are only using one year of SMOS observations. Also, they report a higher sensitivity to AGB at L-band than at higher frequencies (K/X/C-bands), but they do not present a clear comparison of the different data sets. Therefore, I recommend this manuscript for publication after addressing the following issues:

We thank the referee for his/her constructive comments. We will add a discussion on how results would change if a different year is used for the study (see plots and table in the answer to comment 3). The results remain unchanged, but we agree with the referee, that the presentation and discussion would look more robust adding other years. With respect to VOD from higher frequencies, we develop below in comment 2.

In addition below, we give more details on these comments and we address all the other comments made by the referee.

1. The results should be re-organized in a more clear and structured way to facilitate readability and comprehension. There are many general references to relevant results in supplementary material that should directly point to a specific figure or table and be commented in the main text. Some choices made in the analysis and presentation of results are unclear (e.g. the stratification per land cover in two biomes should be further justified) and it is hard to follow the results presented in the main and the supp. material.

Taking into account the three reviewers comments (Referee #2 thinks that "too much importance is given to the inter-comparison of different VOD retrieval algorithms), we reckon that the best trade off is to leave in the supplement the results for SMOS L2 and L3 but to move to the main body of the analysis per biomes currently in supplementary (text and Fig. S6). All land cover classes will be shown separately and Fig. 3 will be removed (see also answer to comment 29). Following this and other related comments, all references to supplementary information in the

main body of the manuscript have been checked and they will be developed more explicitly in a new revised version.

2.  It is unclear how the authors obtain the results plotted in Fig.  4.  It seems they do not use K/X/C-VOD data from year 2011 for a fair comparison to the presented results with L-band and NDVI. Instead, they show the results from Liu et al 2015, which are based on VOD time series from 1993 to 2012 and a  significantly different approach. I believe the data is not directly comparable and the result presented in the figure is therefore misleading. I strongly suggest the authors to either a) include the K/X/C-VOD data from the same study period (yearly average) and detail in the methods or b) focus on the comparison of L-band VOD and NDVI and AGB. I would particularly encourage the latter. Also, the results on Fig 4 could be shown for the four different AGB data sets used in the study, for completeness.

This manuscript is devoted to SMOS L-Band VOD. That's the reason why we did not attempt to perform a complementary study with data from other radiometers at the present stage. However, we do think that it is interesting to discuss the new results by comparing to previous results reported in the literature for other frequency bands. Figure 4 does not contain any new result. It is a Figure for the discussion, where results presented earlier are compared to published results by Liu et al. 2015. However, we realized that the normalization used to plot L-VOD,  K/X/C-VOD and NDVI in the same plot could be misleading. The normalization is not needed to compare with other VOD, only for NDVI. Therefore, we will present the results in a new figure with two panels as the figure below. In the left panel L-VOD and NDVI were normalized to 1 using their maximum values. This is needed to plot the two quantities in the same figure. In the right panel, L-VOD and K/X/C-VOD relationship to Saatchi AGB are shown without using any normalization. The curves plotted here for the K/X/C-VOD are just those of Figure S4 from Liu et al. 2015, which were computed using Saatchi AGB and the same method that we used in the current study. Liu et al fitted their relationship using  K/X/C-VOD data in the period 1998-2002 and Saatchi data acquired from 1995 to 2005 (page 6 of their supplementary information document). This will be reminded explicitly in the discussion section of a revised version of the manuscript. However, the non-linearity of the curve and the difference sensitivity to high AGB from different frequencies is driven by the high AGB values in the dense equatorial forest, which is not supposed to vary strongly in a few years time.

The curves for the other AGB dataset with respect to L-VOD are already shown in Fig S3. They will not add much information to this discussion and we tried to show them in the figure below but it becomes unreadable.

[Figure]

[Figure]

**Figure 5.** Left: Fits of the 5th and 95th percentile curves of the Saatchi et al. (2011) AGB with respect to SMOS-IC L-VOD (green) and NDVI (pink). To plot both distributions with the same scale, VOD and NDVI were normalized from 0 to 1 using their respective maxima (0.83 for NDVI and 1.24 for L-VOD). Right: Fits of the 5th and 95th percentile curves of the Saatchi et al. (2011) AGB with respect to SMOS-IC L-VOD (green) overlaid in the K/X/C-VOD versus Saatchi et al. (2011) AGB curves of Fig. S4 from Liu et al. (2015) (brown). No normalization is needed in this case as both VODs span a similar range of values.

Following reviewers comments, the text on Sect. 4.4 discussing this figure will be moved to Sect. 5 "Discussion" and will add the more detailed explanations provided here-above.

3. The title is too ambitious and general. The focus is clearly on SMOS L-band VOD and biomass, but the results presented (using 1 year of observations over Africa) do not support the use of the words "high sensitivity". I would recommend the authors to provide a more specific title, more representative of its contents.

Taking into account comments from Referee #2 as well, we decided to change the title to:

*An evaluation of SMOS L-band vegetation optical depth (L-VOD) data sets: a high sensitivity of L-VOD to above-ground biomass in Africa*

Otherwise, by "high sensitivity" we meant that the AGB and L-VOD relationship is smooth and with a moderately low slope. This is not related to the number of years used for the study. However, as already mentioned, in the corrected version we will show that using more years do not change this result. For instance, the figure below will replace Fig. S3. Both are almost indistinguishable, however the figure below has been computed using data from two years (2011 and 2012) and both ascending and descending orbits (taking into account the comment on the orbits below).

See also answer to comment 19 below.

[Figure]

4. Section 5 "Discussion" is too short. Results are already discussed in Section 4, and Section 5 adds a brief overview and a comparison to literature studies. I would recommend the authors to re-organize the manuscript and include the content of Section 5 either in the results or in the conclusions as "Discussion and Conclusion".

The referee is right that there are a few comments on results from the literature already in Section 4 "Results". They will be moved to section 5 "Discussion". In addition section 4 will be enlarged with the discussion of the results by biomes following the suggestions (here below) of the reviewer.

Here is a list of more specific comments and recommendations:

1. Abstract, last sentence. Consider changing "index" by "indicator"

We agree. This will be changed.

2. Page 2, lines 9-11. In presence of vegetation, part of the soil emission is absorbed and scattered.  There are two microwave vegetation parameters that are used in the physical model to account for the effect of vegetation: the vegetation optical depth and the single effective scattering albedo.  The authors should introduce here the albedo parameter, or at least mention it.

Thanks for pointing this out. We agree that the best would be to introduce the tau-omega model already here. This will be done as follows. Instead of (lines 10-14):

*In the presence of vegetation, part of the soil emission is absorbed and scattered. This extinction effect is parameterized by the vegetation optical depth (VOD) that can be estimated using radiative transfer theory [...] Wigneron et al. 2007).*

It will be rephrased to:

*In the presence of vegetation, part of the soil emission is absorbed and scattered. These effects can be parameterized using radiative transfer models such as the so-called tau-omega model (Refs), were tau is the optical depth and omega is the single scattering albedo. Tau was shown to be linked [...] Wigneron et al. 2007). Therefore, tau is commonly known as Vegetation Optical Depth (VOD).*

3.  Page 2, line 16.  Specify how "thick" is the vegetation layer that microwaves penetrate, and introduce here a comparison between frequencies (this is later briefly discussed in line 30).  Is the soil emission from tropical and boreal forests reaching the satellites operating at C/X/L bands?  Add references and a brief discussion to support and clarify how the different frequencies are complementary.

*"Thick"* will be removed (*VOD samples the vegetation including the woody vegetation under the green canopy*) as it is difficult to quantify it. We prefer to say that it is thicker than the layer sampled at higher frequencies as done in Line 30. Otherwise, the goal of this paragraph was to cite some examples of studies of the vegetation with VOD. The actual comparison of frequencies is done in the next paragraph (line 23 onwards) and in the first paragraph of page 3.

4.   Page 3,  first paragraph.   Literature on SMAP L-band VOD is totally missing and should be added. For instance, a global comparison of SMAP VOD to lidar-based vegetation height is reported in Konings et al.  2017.  A.G. Konings, M. Piles, N. Das, D.Entekhabi, L-Band Vegetation Optical Depth and Effective  scattering Albedo Estimation from SMAP, Remote Sensing of the Environment, Vol. 198, pp 460-470, 2017.

This is a pertinent paper that will be added to the introduction, together with Konings, A. G.; Piles, M.; Rötzer, K.; McColl, K. A.; Chan, S. K. & Entekhabi, D. Vegetation optical depth and scattering albedo retrieval using time series of dualpolarized L-band radiometer observations Remote Sensing of Environment, 2016, 172, 178-189

5.  Page 3,  line 17.  It should be relevant to (at least) mention briefly the difference between active and passive microwave sensing of vegetation.

Line 18 will be continued as follows:

*[…] observations. In contrast to passive measurements, for which the goal is study how the thermal emission arising from the Earth is affected by the vegetation layer, active measurements allow to study how the radiation emitted by a human-made radiation source is backscattered by the vegetation, which depend mainly in vegetation water content and the vegetation structure.*

6. Page 3, line 25. Please, add a reference to support that the quality of the ascending data is better than the descending. I would "a priori" recommend to use both to increase coverage.

Ascending orbits data have been shown to give somewhat better results than descending orbits to retrieve soil moisture in Europe, North America and the Sahel (see Kerr et al. 2016, RSE, and references therein). The reason is that in some regions they can be less affected by radio frequency interference and that at 6 AM (ascending orbits) the soil and canopy are closer to thermal equilibrium and there are less problems of convective precipitations than for descending orbits (6 PM). However, for a sensitivity study of VOD to vegetation and in particular biomass this does not necessarily apply.

See also answer to general comment 3 above and comment 19 below. We show that the results obtained using descending orbits are same as those obtained using ascending orbits.

In a revised version of the manuscript we will use two years of data (2011-2012 and both ascending and descending orbits) as mentioned in the answer to general comment 3.

7.  Page 4, line 7.  SMOS is first introduced as a full-polarization radiometer but here it is stated that only dual-polarization measurements are used in the retrievals. Why? Too much information to constrain retrievals? Consider including a reference here.

The parameters Stokes 3 and 4 are actually used for filtering the SMOS brightness temperatures, for instance to detect RFI (Kerr et al. 2012, TGRS).  This will be added to the text of the revised version.

8.  Page 4, line 14.  The authors mention that previous L-VOD retrievals are used to constrain new retrievals. How many closest retrievals? Please, be more specific.

Due to the specificities of the SMOS geometry of observation, the profiles of brightness temperatures observed at the middle part of the field of view (~600 km centered on the satellite sub-track) have larger ranges of incidence angle than the outer pat of the field of view. For such observations, the retrieval system has more information content to discriminate the vegetation emission from the ground emission leading to more accurate retrieved soil moisture and VOD. The retrieved VODs and associated uncertainties for such grid points are used as prior first guess and uncertainties for the VOD retrieval of the next overpass of these grid-points (3 days later max) that will be observed, this time, at the outer part of the field of view with reduced range of incidence angle instead of using auxiliary data LAI, LAImax as first-guess values and fixed large prior uncertainties (see Kerr et al 2012).

This information will be added to page 4.

9.  Table S1.  It would be relevant to include how albedo and soil roughness are computed in the different products. Also, please detail previous retrievals. ISEA should be ISEA4h9.

A) **Albedo and roughness:**

For SMOS IC the roughness and single scattering parameters are assigned per IBGP classes, based on Parrens et al. (2017a, b), and are averaged per pixel according to the fraction of classes present in the pixel (Fernandez-Moran et al. 2017).

For SMOS L2 and L3 algorithms, single scattering albedo and roughness values depend on the surface type and are taken from literature and/or specific SMOS studies. For low vegetated area the single scattering albedo is set to 0 and roughness set to 0.1. For forested areas the single scattering albedo is set to 0.06 for tropical and subtropical forest and 0.08 for Boreal forest and roughness set to 0.3 (Rahmoune et al.  2013,2014, Al Bitar et al. 2017).

Marie Parrens, Jean-Pierre Wigneron, Philippe Richaume, Ahmad Al Bitar, Arnaud Mialon, Roberto Fernandez-Moran, Amen Al-Yaari, Peggy O'Neill, and Yann Kerr, 2017. Considering Combined or Separated Roughness and Vegetation Effects in Soil Moisture Retrievals, International Journal of Applied Earth Observation and Geoinformation 55, 73-86.

M. Parrens, A. Al Bitar, A. Mialon, R. Fernandez-Moran, J.-P. Wigneron, P. Ferrazzoli, and Y. Kerr, 2017. Estimation of the L-band Effective Scattering Albedo of Tropical Forests using SMOS observations, IEEE GRS Letters 2017, 14, 1223-1227

b) **Previous retrievals:** see answer to the next comment #10.

c) **Grid name:** The name of the grid will be corrected to reflect the exact name.

10. Page 5, line 6. Mention how SMOS-IC is initialized and refer to Table S1.

In a first run the minimization is initialized with SM 0.2 m3/m3 and L-VOD 0.5. This allowed to compute a mean L-VOD map per each grid point. In a second run  SM was initialized at 0.2  m3/m3 while the mean L-VOD for each grid point was used to initialize the L-VOD. This information will be added to page 5 referring the Table S1 where it was already given but with a typo as the value quoted for the initialization of VOD in the first run was 0.2 instead of 0.5.

11. Page 5, line17. A reference is needed for Worldclim data.

Absolutely, thanks for pointing this out. The reference is: Fick, S. E. & Hijmans, R. J. WorldClim 2: new 1-km spatial resolution climate surfaces for global land areas *International Journal of Climatology, Wiley Online Library,* 2017

12. Page 5, line 21. change "sential" by "essential" (?)

Thanks for the typo correction.

13.  Page 5, line 24.  Consider adding a refernce for EVI and its main differences to NDVI.

The following text will be added with an explicit reference. First, Huete et al. 2002, cited in the first line of the subsection for NDVI will be replaced by:

Tucker, C. J. Red and photographic infrared linear combinations for monitoring vegetation *Remote sensing of Environment, Elsevier,* **1979***, 8*, 127-150

Second, the EVI description text will be enlarged with:

*The enhanced vegetation index (EVI) was designed to have a higher sensitivity in high biomass regions than NDVI by allowing to distinguish the vegetation and the atmosphere contributions to the signal (Huete et al. 2002).* Whereas the (NDVI) is chlorophyll sensitive, *the EVI is more responsive to the canopy type and structure (including LAI) and, for example, it has allowed to study the Amazon green-up season (where other vegetation indexes such as NDVI do not show any particular pattern, Huete et al. 2006).*

Huete, A. R.; Didan, K.; Shimabukuro, Y. E.; Ratana, P.; Saleska, S. R.; Hutyra, L. R.; Yang, W.; Nemani, R. R. & Myneni, R. Amazon rainforests green-up with sunlight in dry season *Geophysical research letters, Wiley Online Library,* **2006***, 33*

14. Page 5, line 7. Words "In a second step" are used in lines 5 and 7.

We assume it is page 6. The second "in a second step" will be replaced by "In the third step" and "in the third step" (line 8) will be replaced by "Finally".

15.  Page7, lines 15-16.  It seems here that the authors hypothesize the AGB data set derived from L-band SAR is probably the more appropriate and therefore they restrict the study to its coverage (i.e. Africa). However, best results are obtained with Saatchi. The authors should better elaborate on why it is important to use this data set and reformulate this sentence.

One cannot summarize the results of this study saying that the best agreement of L-VOD is found with respect to Saatchi AGB dataset. Please, see the answers to comments 26 and 29 here below. Regarding the sentence referred to here, it will be reformulated as : [...] *because one of the reference data-set is available only in Africa, in addition this dataset is particularly interesting because it has been obtained using radar observations at a lower frequency than other datasets, namely, in L-band, which is also the frequency of SMOS. The African continent contains [...]*

16. Page 7, line 24. Please, specify which parameter is used to select the lower values of the cost function (chi-square?)

Yes, Chi2.

17.  Page 7, line 28.  There are different criteria to filter out the quality of SMOS observations. As a common practice, the DQX parameter is used. However, the authors here propose to use the Chi2 parameter larger than 3. A reference should be added to support this criteria.

The DQX is actually a standard deviation which informs only about the uncertainty of the retrieved solution which is driven by the forward model sensitivity at the solution point. It is the retrieved parameter post uncertainty computed using the inverse linear tangent model (Jacobian) at the solution used to translate the observation uncertainty (radiometric accuracy) into the parameter space uncertainty. It does not inform by itself about the correctness of the solution with is based on a quality of a fit. In other words, we can have a very wrong modeling (bad fit) with a retrieved parameters solution where the forward model is very sensitive resulting to low DQX. Moreover, the DQX values are not homoscedastic as our forward models are much more sensitive for lower values of the (SM,VOD) parameter space (leading to low DQX) than for higher values (leading to high DQX). By filtering the DQX too strictly there is a serious risk to bias statistics toward lower retrieved SM and VOD, which would bias our results for tropical forest where both SM and VOD are high.

The DQX should be used as a weight in the parameters use e.g. as it is done by assimilation system. See for instance: A. Tarantola; Inverse Problem Theory and Methods for Model Parameter Estimation, SIAM, 2005.

In contrast, the Chi2, or alternatively its probability, which is naturally used in the retrieval procedure is currently the preferred option to filter out the retrieved solution; it is the classical goodness-of-fit test. See for instance Román-Gascon et al 2017 (using Chi2 < 3.5) or Bircher et al. (2013), who used Chi2 probability.

Román-Cascón, Carlos, et al. "Correcting satellite-based precipitation products through SMOS soil moisture data assimilation in two land-surface models of different complexity: API and SURFEX." *Remote Sensing of Environment* 200 (2017): 295-310.

Validation of SMOS L1C and L2 Products and Important Parameters of the Retrieval Algorithm in the Skjern River Catchment, Western Denmark, IEEE Transactions on Geoscience and Remote Sensing, pp 2969 – 2985. S. Bircher, N. Skou, Y. H. Kerr, 2013, DOI  - 10.1109/TGRS.2012.2215041, Vol 51, Issue 5, ssn 0196-2892

The manuscript text will be modified as follows:

*Several quality indicators are present in the SMOS products. The DQX parameter uses the inverse linear tangent model (Jacobian) to translate the observation uncertainty (radiometric accuracy) into the parameter space uncertainty. The forward models are much more sensitive for lower values of the (SM,VOD) parameter space (leading to low DQX) than for higher values (leading to high DQX). Therefore, filtering using DQX implies a risk to bias our results for tropical forest where both SM and VOD are high. In addition, the DQX parameter does not give information about the correctness of the solution, which is based on a quality of a fit. Therefore, the Chi2 parameter (goodness of the fit) was used to filter out the retrieved solutions. Several tests were done and a value of 3, corresponding approximately to the peak of the Chi2 probability distribution was found to be a good threshold. This is in agreement with the values used in other studies (see for instance, Roman-Cascon et al. 2017).*

18.  Page 7, line 30.  It would be important to show a map with the final number of samples used per pixel, after the filtering criteria is applied. It would also be relevant to show a map of the standard deviation of the estimates (apart from the average on Fig. 1).  This is critical, since the study is based on a final comparison of spatial maps.

In a corrected version of the manuscript we will split Fig 1 in two different figures: one for maps that have been averaged on time and another one for AGB and cumulated precipitations datasets. The first one will add the STD and the number of points in the times series for each grid point as follows.

[Figure]

19.  Page 8, line 3.  The authors average on a yearly basis since they chose only one year of observations. A seasonal study would be interesting, but of coarse more years would be needed. The choice of using only one year of SMOS observations should be further justified.  Also, the impact of using one year in the results should be (at least) discussed later in the manuscript.

Actually, the shape of the scatter plots is very similar using data for other years. See for instance the next figure and compare to Fig S3.

[Figure]

[Figure]

Fig. Left : 2012 Ascending orbits. Right: 2012 Descending orbits

The next table shows the parameters of the fits using ascending or descending orbits in 2012. The values can be compared to those of Table S2. They have almost the same numeric values.

| AGB | Year | Orbit | curve | $a$ | $b$ | $c$ | $d$ | $R^2$ |
|-----|------|-------|-------|-----|-----|-----|-----|-------|
| Avitabile | 2012 | A | 05th | 262.697 | 14.406 | 0.838 | 5.347 | 0.99548 |
| Avitabile | 2012 | A | Mean | 370.615 | 8.920 | 0.727 | 4.915 | 0.99917 |
| Avitabile | 2012 | A | 95th | 462.962 | 9.398 | 0.580 | 1.613 | 0.99243 |
| Saatchi | 2012 | A | 05th | 333.496 | 4.642 | 0.906 | -3.338 | 0.99080 |
| Saatchi | 2012 | A | Mean | 280.675 | 6.686 | 0.684 | 14.439 | 0.99334 |
| Saatchi | 2012 | A | 95th | 291.444 | 9.661 | 0.545 | 32.436 | 0.99372 |
| Baccini | 2012 | A | 05th | 497.748 | 2.673 | 1.024 | -41.748 | 0.98810 |
| Baccini | 2012 | A | Mean | 414.033 | 3.503 | 0.717 | -27.346 | 0.99882 |
| Baccini | 2012 | A | 95th | 383.823 | 4.950 | 0.550 | -1.911 | 0.99757 |
| Bouvet-Mermoz | 2012 | A | 05th | 292.291 | 4.800 | 0.963 | 4.717 | 0.97445 |
| Bouvet-Mermoz | 2012 | A | Mean | 319.123 | 5.261 | 0.760 | 7.879 | 0.99616 |
| Bouvet-Mermoz | 2012 | A | 95th | 358.172 | 7.290 | 0.581 | 17.621 | 0.99226 |
| Avitabile | 2012 | D | 05th | 264.867 | 13.984 | 0.833 | 5.353 | 0.99548 |
| Avitabile | 2012 | D | Mean | 368.991 | 9.271 | 0.726 | 5.795 | 0.99765 |
| Avitabile | 2012 | D | 95th | 483.476 | 7.893 | 0.614 | -2.245 | 0.99585 |
| Saatchi | 2012 | D | 05th | 332.540 | 4.976 | 0.890 | -5.476 | 0.99111 |
| Saatchi | 2012 | D | Mean | 287.547 | 6.436 | 0.691 | 12.981 | 0.99499 |
| Saatchi | 2012 | D | 95th | 291.493 | 8.979 | 0.543 | 30.449 | 0.99324 |
| Baccini | 2012 | D | 05th | 606.345 | 2.250 | 1.168 | -53.195 | 0.98742 |
| Baccini | 2012 | D | Mean | 398.354 | 3.688 | 0.706 | -21.945 | 0.99806 |
| Baccini | 2012 | D | 95th | 379.008 | 5.076 | 0.567 | 6.162 | 0.99538 |
| Bouvet-Mermoz | 2012 | D | 05th | 308.430 | 4.441 | 0.965 | 0.250 | 0.98165 |
| Bouvet-Mermoz | 2012 | D | Mean | 317.532 | 5.425 | 0.764 | 10.022 | 0.99729 |
| Bouvet-Mermoz | 2012 | D | 95th | 375.791 | 6.420 | 0.604 | 12.619 | 0.99451 |

Therefore, there is no real impact of using just one year. However, in a revised version of the manuscript we will use two years of data both for ascending and descending orbits. The results will not change but they would look more robust to the reader. This will be discussed in a revised version of the manuscript. See also answer to General Comment 3.

21. Page 4, line 15. It would be relevant to detail the function used for the fitting in the main manuscript.

We think that the reviewer meant "Page 8" and his/her comment refers to Liu et al. 2015 function.

$$AGB = a \times \frac{\arctan(b\,(\mathrm{VOD} - c)) - \arctan(-bc))}{(\arctan(\infty) - \arctan(-bc)} + d,$$

And the equation of the logistic function. Eq. S3. Of course, both can be shown in the main manuscript.

22. Page 8, line 22. Please, detail "the remaining static data sets" and comment on Figure 1 (e.g. main visual differences between the VOD products and the AGB ones)

"Remaining static data sets" will be changed by "other evaluation data sets". The following description will be added to a revised version of the manuscript. See the figure in the answer to comment 18, which will be the new Fig. 1, the remaining panels (evaluation datasets) of former Fig. 1 will become Fig. 2 in the corrected manuscript.

*Figure 1 shows the average L-VOD computed over 2011 and 2012 using both ascending and descending orbits for the three SMOS L-VOD products. In addition, it also shows the standard deviation (STD) and the number of points of the local time series after applying the filters discussed in Sect 3. The three SMOS L-VOD products show a similar spatial distribution but the SMOS-IC L-VOD shows a smoother spatial distribution than the L2 and L3 datasets. The highest values are found in equatorial forest regions and L-VOD decreases monotonically with distance to the equatorial forest in the tropical area and beyond. The STD of the L-VOD time series also increases towards the equatorial forest, in particular for the L2 and L3 datasets. The number of points in the time series is lower for the IC dataset due to the lower revisit frequency arising from the requirement of having brightness temperature measurements spanning an incidence angle range of at least 20º (Fernandez-Moran et al. 2017).*

Figure 2 shows the evaluation data *after resampling to a 25 km EASEv2 grid*: 2011-2012 average *of the MODIS NDVI and EVI indices, tree height, cumulated precipitations and AGB datasets. EVI and NDVI also decrease with increasing distance to the equator but more slowly than L-VOD. The tree height map shows two main populations: the equatorial forest, with heights larger than 20 meters, and the rest of the continent, where most of the vegetation is lower than 4 meters. In contrast to the previous quantities, AGB can vary in two orders of magnitude, therefore AGB maps are shown in logarithmic units in Figs 1. The Baccini, Saatchi and Bouvet-Mermoz maps show a similar AGB distribution. In contrast, the Avitabile*

*map shows a much sharper decrease of AGB from the equatorial forest region to the rest of the continent.*

23. Figure 1. The reference to Mermoz is missing.

Thanks, it will be added to a corrected version.

24.  Page 9, line 1.  Comment on Spearman and Kendall results, which confirm the results obtained with Pearson.

The referee is right that the table contains all values while the text commented only on Pearson. As he/she says, the Spearman and Kendall results confirm the Pearson results, which also means that the lower values obtained for the L3 dataset are not due to a correlation that could be good but more non-linear than those of the IC and the L2 dataset. Thus, we fully agree that the results should be commented. We propose the following rewriting:

*A quantitative assessment of the correlation and the dispersion of the different scatter plots can be found in Table 1, where Pearson, Spearman and Kendall correlation coefficients are given for the three L-VOD data sets with respect to the evaluation data sets. The lowest Pearson correlation coefficient values were obtained for L3 L-VOD (R = 0.65−0.87). The Pearson correlation coefficients obtained for L2 L-VOD are similar (R = 0.67−0.87) than those obtained for L3 L-VOD but systematically higher by up to 4%, while the values obtained for IC L-VOD are the highest (R = 0.77−0.94) with respect to all the evaluation data sets. The correlation increase is in the range of 5%-10% with respect to L2 L-VOD and up to 15 % with respect to L3 L-VOD. The rank correlation values with respect to all the evaluation datasets are also higher for IC L-VOD (rho 0.78-0.91, tau 0.61-0.75), followed by L2 L-VOD (rho 0.67-0.83, tau 0.50-0.65) and L3 L-VOD (rho 0.66-0.80, tau 0.49-0.62). These results are in agreement with those obtained with the Pearson correlation and imply that the lower Pearson correlation values obtained for the L3 and L2 datasets are not due to a correlation that could be better but more non-linear than that of the IC dataset. Therefore, using eight vegetation-related evaluation data sets and three different metrics, the most consistent SMOS L-VOD data set is SMOS-IC. This result implies that, currently, the SMOS-IC dataset is the best SMOS L-VOD product to perform vegetation studies, and the rest of the current study will focus on SMOS-IC L-VOD.*

25. Page 9, line 20. It is interesting that only with Saatchi and Baccini there is a single AGB peak corresponding to the higher VOD values.  Why do the authors believe this peak is not appearing as clearly with the other two data sets?  Is it consistent that the peak is higher for Baccini than for Saatchi? The authors should elaborate on the results presented.

We have analyzed the high AGB blobs of the scatter plots as follows:

Avitabile blob 1:  VOD > 1 ; 230 < AGB < 330

Avitabile blob 2: VOD > 1 ;  AGB > 330

Bouvet-Mermoz blob 1: VOD > 1 ; 170  < AGB < 260

Bouvet-Mermoz blob 2: VOD > 1 ; AGB > 270

Saatchi : VOD > 1; AGB > 240;

Baccini: VOD > 1; AGB > 240;

The next figure shows the spatial distribution of those peaks:

[Figure]

In the two upper rows, one sees that consistently for Bouvet-Mermoz and Avitabile the first blob (slightly lower AGB) is in the center of the equatorial region around the Congo river basin while the spatial distribution for the highest AGB blob surrounds the first one. This bi-modal behavior is not seen for the high AGB values in the Saatchi and Baccini datasets, where the whole equatorial forest shows more homogeneous distribution with similar values in the two regions. Definitely, L-VOD seems to be in more agreement with the two latter datasets, unless the high AGB blobs in Bouvet-Mermoz and Avitabile are more realistic and in this case, L-VOD would show signs of saturation since, it remains basically constant. This is the same discussion already done when commenting the scatter plots. Thus, we reckon that it is not necessary to add anything to a revised version of the manuscript. In any case, to our knowledge, it is not easy to say which of the four AGB datasets is more realistic in the densest parts of the equatorial forest.

Note that the spatial distributions shown in the previous figures are not an artifact arising from the spatial averaging of the AGB to the SMOS resolution. If one plots the AGB data at the original resolution the differences are clear. See for instance in the next two figures that the Baccini original data (upper figure) is much more homogeneous than the Avitabile data (lower figure).

[Figure]

[Figure]

26. Page 9, line 22. It seems to me that also Saatchi shows a very low dispersion for low AGB values, but the plot is too small. Please, address.

That's true. Together with Bouvet-Mermoz, the Saatchi dataset show the highest correlation values with respect to SMOS L-VOD. In Fig. S6, one can see that correlation coefficients obtained with Saatchi's data are somewhat higher than those of Bouvet-Mermoz for low vegetation but lower for Savannahs. For woody savannah the situation is more complex for the mixed nature of this biome and because in the Bouvet-Mermoz datasets uses the Mermoz law for pixels classified as dense forest on the ESA CCI land cover dataset. Pixels classified as woody savannah using IGBP at the SMOS resolution can contain both woody savannnah and dense forest in the ESA CCI dataset. The scatterplot for woody savannah using the Bouvet-Mermoz dataset show these two populations for high L-VOD values, which decreases the Pearson correlation (which is lower than that obtained for Saatchi in woody Savannah) but still, Kendall and Spearman correlations are higher for the Bouvet-Mermoz dataset. This discussion will be added to a corrected version of the manuscript. See also answer to comment 29.

27. Page 9, line 29. The authors aggregate the data sets in two groups of biomes. This separation should be further justified. Also, there are many results shown in the supplementary material that are relevant and should at least be discussed in the text.

We understand that the referee is suggesting to move Fig. S6 and the text in page 2 Lines 16-31 to the main body of the manuscript. This seems a good idea for a revised version of the manuscript. Of course, the additional discussion suggested in the previous point will also be added.

28.  Section 4.9.  I would suggest the authors to include a box plot with the SMOS IC VOD results per land cover.  It will give a general idea of the dispersion and the mean values of VOD per land cover. Perhaps it would also be good to show the box plots for the AGB data sets.

There is not section 4.9 and it is not fully clear to us what the reviewer is suggestion. Making box plots showing the distribution of LVOD for different land cover classes? We recknon that this will not add much information as the distribution can be seen also in the scatterplots per land cover class. Maybe he/she is proposing something else ?

29. Section 5. It would be nice to add a discussion on the consistency of the four AGB data sets and on why best correspondence is found between L-VOD and the approach of Saatchi (and not the one of L-band SAR).

We do not think that one can summarize the results of this study saying that the correspondence of L-VOD and AGB is better with Saatchi AGB. Figure S6 clearly shows this. For instance for woody savannah the Pearson correlation is higher for Baccini and all three correlation coefficients are also higher for Baccini AGB in dense forest (see also the answer to comment 26). In any case, we will proceed as indicated in the answer to comment 27 and we will move Fig. S6 to the main body showing separately all the classes (separating shrublands, crops, grasslands and natural vegetation, see the two figures below). The best correlations of AGB and L-VOD are found with (i) Bouvet-Mermoz for Shrublands and Savannahs (ii) Baccini for croplands and equatorial forest (iii) Saatchi for grasslands. Regarding natural vegetation and woody savannah the correlation values obtained with Saatchi and Baccini are very similar. One should note that correlation values obtained with Bouvet-Mermoz for woody savannah are degraded to the fact that for the highest values of AGB found in this class at the SMOS resolution, the AGB estimation is a mix of Bouvet and Mermoz approaches. Therefore, all AGB datasets except that of Avitabile performs the best for a few land cover classes.

L-Band radar observations are thought to be very sensitive to biomass variations, in spite of a significant sensitivity to soil moisture as well. The high correlation with SMOS L-VOD, also at L-band, would confirm this fact. The strange behavior of Avitabile AGB probably comes from the fact that it is pure data driven method and that it is therefore very sensitive to the data used to train the method. In their training database, high AGB plots could be over-represented.

[Figure]

**Figure 4.** SMOS IC L-VOD relationships to the AGB and tree heigh evaluation datasets for different land cover classes: From left to right: *(i)* Open shrublands. *(ii)* Croplands, *(iii)* Grasslands, and *(iv)* Cropland/natural vegetation mosaics. See Table S2 for more details on these land cover classes.

[Figure]

**Figure 5.** SMOS IC L-VOD relationships to the AGB and tree heigh evaluation datasets for different land cover classes: From left to right: *(i)* Savannah *(iii)* Woody savannah and *(iv)* Evergreen broadleaf. See Table S2 for more details on these land cover classes.

30. Figure 3. It would be interesting to know the number of pixels in the two groups of biomes, and whether they are balanced. Are all the correlation significant? To what level? This is important information that should be included either in the figure or the text.

The number of grid points in the two groups of biomes it is, of course, not the same. That is not the point here, the point is that groups from grassland, croplands, savannah and woody savannah show a similar slope, much lower than that of the equatorial forest (Fig. S6, which will be moved to the main text) therefore they can be grouped in two groups because they show two "regimes" of the AGB vs L-VOD relationship. All correlations are significant with very low P-values (<0.05).

31. Page 10, line 3. The authors should comment on the slope of NDVI per land cover and most relevant aspects shown in the supplementary information.

NDVI and EVI will be added to current Fig. S6 and moved to the main body.  Former Fig. S6 will become two new figures to show all land cover classes and Fig. 3 will be removed. The new Figures are shown in the answer to comment 29.

Regarding the slopes of the relationship, the following text will be added to the manuscript.

*Regarding the L-VOD and NDVI/EVI relationships in different biomes, it is worth noting that, in contrast to AGB, the slope of the relationship decreases from low vegetation types to savannahs and dense forest as the optical/infrared indices saturates. It is noteworthy that no significant difference is seen on the behavior of EVI and NDVI for high L-VOD values.*

32.  Page 10,  line 17.   Please,  specify which part of the supplemtary information is being referred to here.

The sentence will be removed as Fig. S6 will be moved to the main document and discussed in detail there.

33. Figure 4. Legend reads "C/X VOD" but caption reads "K/X/C VOD". Please correct.

Thanks for pointing this out. The legend should actually show "K/X/C VOD". See answer to general comment 2 above.

---

## Author Comment (AC2) · 4 May 2018

The paper provides evidence that the vegetation optical depth VOD derived from passive microwave satellite data at L-band frequency has strong correlation with the above-ground biomass and can be used to monitor vegetation status. The paper is well-written and the methodology and results are sound and at the same intriguing, suggesting VOD as a potential satellite derived parameter to explore in future studies. I recommend the paper for publication but I have few suggestions and recommendations that may help improve the interpretation of the results before final publication of the paper.

We thank the reviewer for his/her encouraging comments

1. The paper does not provide a strong motivation of what VOD can be used for. Vegetation aboveground biomass is one of the most important global ecosystem variable for carbon cycle and climate mitigation. However, the strong correlation of VOD with biomass does not necessarily mean VOD from passive microwave at approximately 0.5-degree resolution is useful for biomass estimation or monitoring. VOD can be used to monitor vegetation water content at regional scales given its coarse resolution and frequent observation. I would like to suggest that although the authors correlate the result with biomass, they emphasize the use of VOD for monitoring vegetation water content. Biomass and water content are similar in magnitude with biomass being more static and water content more dynamic.

Examples of the use of VOD for vegetation, in general, and AGB monitoring, in particular, are given in the introduction (page 2 lines 17-23). We do think that VOD, in particular L-VOD is useful to monitor the temporal evolution of AGB at a lower spatial resolution but with a higher temporal resolution than other types of observations at least until the launch of the ESA Biomass mission, whose goal is to produce a global biomass map twice per year. Liu et al. 2015 have provided a very good example on how L-VOD can be used to study the evolution of global carbon stocks. We do think that all this pieces of information are already in the manuscript since the abstract and the introduction.

Regarding the use of VOD to monitor the VWC, the reviewer is right. Everything depends on the temporal scale of the study. The current manuscript being devoted mainly to AGB, long times periods were used. Studying the evolution of VWC requires to use much shorter time scales and it is an on-going work for a dedicated study. We agree that this was not fully explicit in the manuscript. In addition, a few important references were lacking:

Konings, A. G. & Gentine, P. Global variations in ecosystem-scale isohydricity Global change biology, Wiley Online Library, 2017, 23, 891-905

Li, Y.; Guan, K.; Gentine, P.; Konings, A. G.; Meinzer, F. C.; Kimball, J. S.; Xu, X.; Anderegg, W. R.; McDowell, N. G.; Martinez-Vilalta, J. & others Estimating Global Ecosystem Isohydry/Anisohydry Using

Active and Passive Microwave Satellite Data Journal of Geophysical Research: Biogeosciences, Wiley Online Library, 2017

Therefore, we will change the text of the introduction and we will include the following sentences:

*VOD has also been used to study the VWC and variations in ecosystem-scale isohydricity (Konings and Gentine, 2017; Li et al., 2017).*

and

*Since this study is mainly devoted to AGB, long time averages (typically annual) will be used. Studying the evolution of VWC would require to use  much shorter time scales.*

2.  The method says:  "The main evaluation strategy used in this study is to spatially compare L-VOD to the evaluation data set." Although the pixel values are extracted from all the data sets to compare.  However, this is not a spatial analysis because the spatial information almost disappears in the correlation studies.  Unless a specific spatial correlation model was used to capture the pattern.  Some of the vegetation classes are separated that can help with spatial variation of the data sets but again this is only a simple correlation study and does not include spatial analysis of data sets.

We agree. "Spatially" can be misleading. It will be removed.

3.  Figure  2.  The density scatter plots with multiple parameters show that there is a strong relationship between VOD and all the parameters.  Some of the most interesting ones are the optical data and  precipitation showing a strong saturation with respect to VOD suggesting that VOD can be used as a complementary measurement to look at the vegetation.  Wavelength is probably the most powerful aspect of the VOD measurements compared to optical data. If VOD correlated with EVI and NDVI over the entire range,  then the interpretation of VOD could've been more difficult. I recommend the authors discuss this in the paper.

Following different reviewers comments, Fig. S6 will be moved to the main text and it will include all the land cover classes without grouping some of them (which requires making two figures instead of one, see Figures below). In addition, the new version will include EVI and NDVI.

[Figure]

**Figure 4.** SMOS IC L-VOD relationships to the AGB and tree heigh evaluation datasets for different land cover classes: From left to right: Open shrublands. *(ii)* Croplands, *(iii)* Grasslands, and *(iv)* Cropland/natural vegetation mosaics. See Table S2 for more details on these nd cover classes.

[Figure]

**Figure 5.** SMOS IC L-VOD relationships to the AGB and tree heigh evaluation datasets for different land cover classes: From left to right: *(i)* Savannah *(iii)* Woody savannah and *(iv)* Evergreen broadleaf. See Table S2 for more details on these land cover classes.

The complementary of L-VOD with respect to NDVI and EVI is clear as both saturates strongly. In addition, it is interesting to remark that there are no significant

differences in between NDVI and EVI, even if EVI is supposed to be more sensitive than NDVI to high AGB.

Lines 2-3 of page 10, dealing the slopes of the NDVI versus AGB relationships will be rephrased as follows:

*Regarding the L-VOD and NDVI/EVI relationships in different biomes, in contrast to AGB, the slope of the relationship decreases from low vegetation types to savannahs and dense forest as the optical/infrared indices saturates. It is noteworthy that no significant difference is seen on the behavior of EVI and NDVI for high L-VOD values.*

With respect to precipitation. Panels c and d of figure 3 will be now a single Figure. The rest of the panels will be removed as they will be redundant with those of the new figures showed above. Here showing only two groups of biomes is pertinent as we want, as the reviewer says, to show the possible link of precipitation as one of the drivers of the vegetation properties. Thus, we show that there are basically two regimes. In the first one, as precipitation increases the amount of vegetation as traced by AGB maps, VOD or NDVI/EVI increases. In contrast, there is threshold of ~1500 mm/year over which AGB, VOD and NDVI are decoupled from the amount of the annual precipitations.

Taking this into account, lines 13-17 of page 10 will be replaced by :

*Regarding the relationship with respect to annual precipitation, Fig. 7 shows the precipitations and L-VOD scatter plots for two land cover groups: (i) grasslands, croplands, shrublands, savannahs and woody savannahs and (ii) ever-green broadleaf forest. For the first group, L-VOD increases with increasing annual precipitations until a value of ~ 0.7 for ~ 1500 mm. In this range of L-VOD all other vegetation tracers increase as well. For instance, Bouvet-Mermoz and Saatchi's AGB increase up to 85 Mg/h and ~ 100 Mg/h, respectively, and NDVI and DVI increase up to ~ 0.7 and ~ 0.45, respectively (Fig. 6). In contrast, over that threshold of ~ 1500 mm of annual precipitations, which occurs basically in the evergreen broadleaf forest, L-VOD and the other vegetation tracers are not linked to the amount of precipitation.*

4. The relationship between VOD and biomass from different products are interesting. The fact that L-band VOD does not show a clear saturation with biomass may be due to:

a.  1.  At very coarse resolution (40-50 km),  the variations of forest biomass on the landscape is dominated with the landscape heterogeneity.  Larger heterogeneity (e.g. forest/non-forest mixture) will improve the relationship of VOD with biomass. This may mean that the VOD is also co-varying with the vegetation cover.  In fact, the straighter relationship with Baccini data is the artifact of this effect.  Baccini biomass is strongly correlated with MODIS VCF (vegetation continuous field) data

and therefore causes a more linear relationship. Whereas other maps and including the vegetation height from Simard do not show this linear relationship.  There is no reason for VOD and biomass to have a linear relationship. I recommend the authors discuss this point and may even include the MODIS VCF product as a layer similar to NDVI in the mix.

At *all* spatial resolutions the sensor output is a weighted average of the signal within the radiometer footprint or the CCD pixel and the instrument response. Since most of the time it is impossible to deconvolve the instrument response function and the function giving the 2D distribution of the signal, the physical parameters retrieved from the sensor output are "effective" values within that footprint or pixel. They are not independent of the 2D distribution … but two completely different 2D distributions can give the same sensor output.

The suggestion of using MODIS VCF is a valuable one, and we will certainly process that data for subsequent studies. For the present study, for simplicity, we have preferred to compute the forest fraction from ECOCLIMAP because it is provided with some SMOS products such as the L2 and L3 datasets as the "FFO" parameter. FFO is the fraction of forest within the SMOS footprint computed from the 1 km ECOCLIMAP dataset. The following figure shows the SMOS IC L-VOD as a function of the fraction of forest cover from 60% to 100 %. The distribution shows two structures. In a first structure, the VOD varies from 0.2 to ~0.7 for those FFO values, showing a very small sensitivity to the exact value of the fraction of forest (small slope) because VOD does not only depend on the fraction of forest but also on the type and the properties of the vegetation within that fraction (and on the low vegetation present outside the forested cover fraction). For instance, it is possible to have almost 0.6 of L-VOD with FFO of 60 % and values as "low" as 0.4 for FFO of 100 %. The second structure also illustrates this effect, it is the high L-VOD (>1) peak for FFO 90%-100%. This is the same peak seen in the scatterplots with respect AGB and that corresponds to the equatorial forest, for which the L-VOD is higher than for other types of forest with the same cover fraction.

[Figure]

We agree that there is no reason to have a linear VOD-AGB relationship and the new version of the figures clearly shows that analyzing the VOD-Baccini AGB per land cover class, the slopes change significantly. The reason that the global relationship looks close to linear is the high slope of the VOD-Baccini AGB relationship in woody Savannahs, which is close to that in evergreen broadleaf forest. With respect to the other three AGB datasets, the Baccini AGB dataset seems to overestimate AGB in woody savannahs with values of ~ 50 Mg/h for LVOD 0.2 to ~130 Mg/h for LVOD 0.6, while the regression lines for the other datasets show values lower than 100 Mg/h. However, unfortunately, the origin of this possible overestimation is not clear and getting further insight on the differences in between Saatchi, Baccini, Avitabile and Bouvet-Mermoz datasets is beyond the scope of the current study.

b. At coarse resolution, the global biomass values are much smaller on the average. Biomass at 1-ha can reach a very large number at some ecosystems. However, at 40 km as it is mixed with the heterogeneity the average is almost smaller. This is one more reason for better sensitivity to biomass. However, it would be interesting to focus on different range of biomass with VOD.

We refer to the answer to comment 4.a.1 regarding the sensitivity to biomass and heterogeneity.

Regarding the analysis for different ranges of biomass and VOD, we refer to the answer to comment 3 and to the discussion of the new figures that will replace Fig. S6. As now all biomes are treated independently, all different ranges of biomass and VOD are discussed independently.

c.  Over Africa,  all dense tropical forests are clustered around 300 Mg/ha of biomass on the graphs in figure 2.  If the goal of the paper is sensitivity to biomass, it may not be a bad idea to separate areas of up to 150 Mg/ha that includes the first cluster from the second cluster and study it separately.  The binary feature of biomass in Africa, from woodlands to dense humid tropical forests in area may introduce a false strong correlation with biomass that need to be discovered further. Figure 3 is supposed to show this effect.  However, the authors mix this up with precipitation and NDVI and only show the result from Bouvet. It would be good to show this for all biomass maps so the variations of the relationships are discussed.

The referee is fully right, Figure 3 and Figure S6 were supposed to show that effect. Taking into account some comments from several reviewers, and as stated in the answers  to comments  3  and 4.b, the former Fig. S6 will be moved to the main document and "expanded" into the two new figures shown above to deal with all biomes independently while former Fig. 3 will be dedicated only to precipitations.

We are confident that the new version with these figures is clearer and it address the concern raised here by the referee. In the new figures, it is possible to see that L-VOD and AGB correlation exist within each land cover class and not only when all classes are shown together.

d. Although the paper is written for the biogeoscience community, it would be important for the authors to provide some explanation of why L-band data from passive measurements may have better relations with biomass compared to active measurements at the same frequency.

Currently we do not have any clear evidence showing that passive L-band data may have better relations with biomass than active L-band data. We do not think we suggest this in the manuscript. The main reason is that all AGB maps used come from active observations. Fully independent AGB estimations as for instance from in situ estimations of AGB would be needed to address that question by comparing to both active and passive L-band observations. But due to the coarse resolution of passive instruments this will be very challenging.

e.  How different are the relationships between VOD and different biomass maps and how can the difference be interpreted?

The best correlations of AGB and L-VOD are found with (i) Bouvet-Mermoz for Shrublands and Savannahs (ii) Baccini for croplands and equatorial forest (iii) Saatchi for grasslands. Regarding natural vegetation and woody savannah the correlation values obtained with Saatchi and Baccini are very similar. One should note that correlation values obtained with Bouvet-Mermoz for woody savannah are degraded to the fact that for the highest values of AGB found in this class, at the SMOS resolution, the AGB estimation is a mix of Bouvet and Mermoz approaches. Therefore, all AGB datasets except that of Avitabile performs the best for a few land cover classes.

Interpreting where do this differences come from is not easy. Radar observations in low vegetation regions such as shrublands and grasslands are thought to be very sensitive to biomass variations, in spite of a significant sensitivity to soil moisture. The high correlation of the two AGB maps mainly based of radar data (Saatchi and Bouvet-Mermoz) with SMOS L-VOD in grasslands would confirm this fact, as the high correlation in shrublands for Bouvet-Mermoz.

 The scatter plot found with Avitabile for woody-savannah resembles an overlay of the scatterplot obtained with Baccini and the scatter plot obtained with Saatchi. The low AGB vs L-VOD slopes obtained for low vegetation classes, significantly lower than those found with the original Saatchi and Baccini datasets, are rather difficult to understand. The strange behavior of Avitabile AGB probably comes from the fact that it is pure data driven method and that it is therefore very sensitive to the data used to train the method. In their training database, high AGB plots could be over-represented.

In addition as mentioned above, the distribution of Baccini AGB for woody savannah is significanly different to the other datasets, which much higher values.

These elements will be added to the results section when discussing the new figures that will replace Fig. 3 and S6.

f.  In table 1, there are three metrics to show the relations between VOD and biomass and other parameters. However, only Baccini result is highlighted in the abstract.  Why?  The table does not necessarily support this.  Furthermore, there is not physical reason that the scattering or emissivity has to be linearly related to biomass.

We understand the point by the reviewer that only Baccini is cited explicitly in the abstract and this could look strange. First, it a good style practice not to make citations in the abstract. In addition, the statement saying that the relationship of Baccini and L-VOD is linear was not correct. It as been removed from the abstract.

*[…] four AGB data sets. The relationships between L-VOD and the AGB data sets were linear per land cover class, but with a changing slope depending on the class type, which makes a global non-linear relationship. In contrast, the relationship*

*linking L-VOD to tree height ($R = 0.87$) was **close** to linear. For low vegetation classes [...]*

Actually, saying that the relationship with respect to Baccini is linear was motivated by the fact that it is closer to linear than those obtained with the other datasets. The degree of non-linearity of the L-VOD/AGB relationship clearly increases from Baccini to Saatchi and Bouvet-Mermoz (which are similar) and to Avitabile, which is strongly non-linear. Finally, we do not reckon that it is needed to give those details in the abstract and that it is better to say that the global relationship is non-linear in all the cases but basically linear per land cover class. The new version of Fig. S6 showing all the land cover classes shows clearly that the relationship of L-VOD and Baccini is not  linear with slopes going from only 2.16-43 Mg/h for shrublands and grasslands to 100-170 for croplands and savannahs and to 220-260 for woody savannah and evergreen broadleaf forest.

Thanks for pointing this out. We agree that there is not physical reason that the scattering or emissivity has to be linearly related to biomass. And furthermore, emissivity is not linearly related to L-VOD either.

5. Figure 4 is a bit difficult to understand. The colors and what the legend provide cannot be easily deciphered. It seems one should the see the saturation of NDVI and a much linear relationship with VOD but I am not sure the figure explicitly shows this.  I recommend either making the figure a bit simple or provide more information in the caption and change colors so the points are clear.

Definitely, this figure is not clear enough as the three referees were concerned about this point. Therefore, we have completely re-thought the best way to present this figure, which basically does not contain any new result and the goal is to illustrate the discussion by comparing L-VOD to results presented earlier and published results by Liu et al. 2015. Therefore, following reviewers comments, and to avoid misunderstandings the text on Sect. 4.4 discussing this figure will be moved to Sect. 5 "Discussion". In addition, the new text will add more  explanations on how the figure was done.

To make a clearer figure we decided to make a new one with two panels (see below).

[Figure]

[Figure]

**Figure 5.** Left: Fits of the 5th and 95th percentile curves of the Saatchi et al. (2011) AGB with respect to SMOS-IC L-VOD (green) and NDVI (pink). To plot both distributions with the same scale, VOD and NDVI were normalized from 0 to 1 using their respective maxima (0.83 for NDVI and 1.24 for L-VOD). Right: Fits of the 5th and 95th percentile curves of the Saatchi et al. (2011) AGB with respect to SMOS-IC L-VOD (green) overlaid in the K/X/C-VOD versus Saatchi et al. (2011) AGB curves of Fig. S4 from Liu et al. (2015) (brown). No normalization is needed in this case as both VODs span a similar range of values.

In the right panel of the new figure, the L-VOD and K/X/C-VOD relationships to Saatchi AGB are shown without using any normalization as we realized that the normalization used to plot L-VOD,  K/X/C-VOD and NDVI in the same plot could be misleading. The normalization is not needed to compare with other VOD, only for NDVI because their dynamic ranges are very different. The curves plotted here for the K/X/C-VOD are just those of Figure S4 from Liu et al. 2015, which were computed using Saatchi AGB and the same method that we used in the current study. Liu et al fitted their relationship using  K/X/C-VOD data in the period 1998-2002 and Saatchi data acquired from 1995 to 2005 (page 6 of their supplementary information document). This will be reminded explicitly in the discussion section of a revised version of the manuscript. However, the non-linearity of the curve and the difference sensitivity to high AGB from different frequencies is driven by the high AGB values in the dense equatorial forest, which is not supposed to vary strongly in a few years time.

In the left panel L-VOD and NDVI relationships with respect to Saatchi AGB. In this case L-VOD and NDVI were normalized to 1 using their maximum values (1.24 and 0.83, respectively)  to have both quantities with the same dynamic range in the same figure. We hope the figure is clearer now. The text will be updated accordingly

The curves for the other AGB datasets with respect to L-VOD are already shown in Fig S3. They will not add much information to this discussion and we tried to show them in the figure below but it becomes unreadable and even more difficult to understand.

---

## Author Comment (AC3) · 7 May 2018

We would like to start thanking the reviewer for his thoughtful and constructive remarks. Please see our response to the specific comments in the attached file. In addition, we will upload a revised manuscript (using track change) and final answers referring to changes in the new version as soon as we are invited to do so by the Editors.

**Referee # 2**

The study is aimed to introduce the sensitivity of the vegetation optical depth (VOD) at L band to the biomass. Different SMOS datasets, produced by different algorithms, are compared to some above ground biomass (AGB) datasets over Africa. The analysis is carried out to show the higher correlation of the L band VOD with respect to higher frequencies VOD and optical vegetation indices. The paper also presents the correlation of the SMOS VOD with other parameters like tree height and cumulated precipitations.

We thank the reviewer for his/her constructive comments.

General comments:

The study's goal is well defined in the paper introduction where the authors claim that the retrieval of the VOD at L band can provide an important tool for the monitoring of the vegetation properties at large scales. In the first section of the manuscript is highlighted that, besides optical measurements, passive microwave observations acquired by the SMOS radiometer can provide an important complementary information to infer the state of vegetation. Here, several references are correctly reported to introduce the study and it is emphasized how the L band observations are less attenuated through the vegetation canopy. Therefore, L band VOD is expected to sample the vegetation layer up to higher biomass values compared to higher frequency observations.  This aspect represents the key point of the manuscript and it is supported by the figure 4 of the results section.  Anyway, just few comments are deserved to this point while a deeper explanation of the high sensitivity of the L band should be provided in the last section of the results.

First, following comments by reviewer #1, we have improved the presentation and the explanation of former Fig. 4.  Sect 4.4 will be moved to the discussion as the description of this figure, in particular using the curves of Fig S4 by Liu et al. 2015 is basically a discussion of new results on L-VOD/AGB with respect to published results by Liu et al. And we would like to avoid misunderstandings on this point. The new figure has two panels. In the left panel L-VOD and NDVI were normalized to 1 using their maximum values. This is needed to plot the two quantities in the same figure. In the right panel, L-VOD and K/X/C-VOD relationship to Saatchi AGB are shown without using any normalization, because they span basically the same range and following comment 2 by reviewer 1, we want to emphasize that the curves plotted here for the K/X/C-VOD are just those of Figure S4 from Liu et al., which were computed using Saatchi AGB and the same method that we used in the current study.  Liu et al fitted their relationship using  K/X/C-VOD data in the period 1998-2002 and Saatchi data acquired from 1995 to 2005 (page 6 of their supplementary information document). The fact that the dates of the different datasets vary will

**Fig. 1.**

---

## Author Comment (AC4) · 7 May 2018

**Referee # 2**

The study is aimed to introduce the sensitivity of the vegetation optical depth (VOD) at L band to the biomass. Different SMOS datasets, produced by different algorithms, are compared to some above ground biomass (AGB) datasets over Africa. The analysis is carried out to show the higher correlation of the L band VOD with respect to higher frequencies VOD and optical vegetation indices. The paper also presents the correlation of the SMOS VOD with other parameters like tree height and cumulated precipitations.

We thank the reviewer for his/her constructive comments.

General comments:

The study's goal is well defined in the paper introduction where the authors claim that the retrieval of the VOD at L band can provide an important tool for the monitoring of the vegetation properties at large scales. In the first section of the manuscript is highlighted that, besides optical measurements, passive microwave observations acquired by the SMOS radiometer can provide an important complementary information to infer the state of vegetation. Here, several references are correctly reported to introduce the study and it is emphasized how the L band observations are less attenuated through the vegetation canopy. Therefore, L band VOD is expected to sample the vegetation layer up to higher biomass values compared to higher frequency observations.  This aspect represents the key point of the manuscript and it is supported by the figure 4 of the results section.  Anyway, just few comments are deserved to this point while a deeper explanation of the high sensitivity of the L band should be provided in the last section of the results.

First, following comments by reviewer #1, we have improved the presentation and the explanation of former Fig. 4.  Sect 4.4 will be moved to the discussion as the description of this figure, in particular using the curves of Fig S4 by Liu et al. 2015 is basically a discussion of new results on L-VOD/AGB with respect to published results by Liu et al. And we would like to avoid misunderstandings on this point. The new figure has two panels. In the left panel L-VOD and NDVI were normalized to 1 using their maximum values. This is needed to plot the two quantities in the same figure. In the right panel, L-VOD and K/X/C-VOD relationship to Saatchi AGB are shown without using any normalization, because they span basically the same range and following comment 2 by reviewer 1, we want to emphasize that the curves plotted here for the K/X/C-VOD are just those of Figure S4 from Liu et al., which were computed using Saatchi AGB and the same method that we used in the current study.  Liu et al fitted their relationship using  K/X/C-VOD data in the period 1998-2002 and Saatchi data acquired from 1995 to 2005 (page 6 of their supplementary information document). The fact that the dates of the different datasets vary will

also be reminded explicitly in the text. We will also remark that the non-linearity of the curve and the difference sensitivity to high AGB from different frequencies is driven by the high AGB values in the dense equatorial forest, which is not supposed to vary strongly in a few years time.

Finally, as suggested by reviewer #2, we will remind that the different shapes of L-VOD vs AGB and K/X/C-VOD vs AGB curves in agreement with what it is expected from the radiation transfer theory (Wigneron et al. 1995, 2004, Ferrazzoli and Guerriero 1996) and previous results on L-VOD and X/C-VOD comparison by Grant et al. 2016 and Vittucci et al. 2015 (already cited in first paragraph of page 3). For instance, the right panel of the figure below clearly shows that for a given AGB, L-VOD is lower than VOD at higher frequencies, as expected. This will be added explictly to the text discussing the new Figure.

[Figure]

**Figure 5.** Left: Fits of the 5th and 95th percentile curves of the Saatchi et al. (2011) AGB with respect to SMOS-IC L-VOD (green) and NDVI (pink). To plot both distributions with the same scale, VOD and NDVI were normalized from 0 to 1 using their respective maxima (0.83 for NDVI and 1.24 for L-VOD). Right: Fits of the 5th and 95th percentile curves of the Saatchi et al. (2011) AGB with respect to SMOS-IC L-VOD (green) overlaid in the K/X/C-VOD versus Saatchi et al. (2011) AGB curves of Fig. S4 from Liu et al. (2015) (brown). No normalization is needed in this case as both VODs span a similar range of values.

Moreover, it seems that the presented research is a progress of a previous work in which some of the authors have already addressed the topic in 2016, including some results about the SMOS VOD sensitivity to tree height and AGB. I would suggest citing also this preliminary study in the introduction (doi 10.1109 / IGARSS.2016.7730383).

This research took, of course, as starting point a literature review and we tried to cite since the introduction all previous relevant studies. For instance, results about the SMOS VOD relationship to tree height were shown by Rahmoune et al. 2014 and Vittucci et al. 2016 (RSE 180). This last paper also included some results with respect to AGB. Both references are cited and commented in the manuscript. The conference contribution cited by the reviewer corresponds to the Vittucci et al. 2016 RSE paper. We reckon that there is not need to cite a conference paper with

preliminary results when the full study has already been published in a peer-review journal.

Another general concern it is related to the use of three different VOD datasets derived from the SMOS data (L2, L3 and SMOS IC) that could confuse the reader. In my opinion, this point of view is interesting but can defocus the attention from the study objective, that it is claimed in the manuscript title. In some parts of the article it seems that too much importance is given to the intercomparison of the different VOD retrieval algorithms, instead of supporting the relevance of the VOD at L band for AGB monitoring. Furthermore, a potential user of SMOS data, could ask himself what is the product to adopt between the L2, L3 and SMOS IC for vegetation monitoring, since the strengths and weaknesses of the different approaches can be highlighted more clearly. A suggestion to address this point could be to provide a general overview of the specific aims of the different products and maybe to update the title of the research to highlight that different L-band products are compared.

**Title:** referee #1 did also think that the title should be changed. Therefore, we decided to change the title to:

*An evaluation of SMOS L-band vegetation optical depth (L-VOD) data sets: a high sensitivity of L-VOD to above-ground biomass in Africa*

**Comparison of SMOS L-VOD products:** we agree completely with the reviewer, that is the reason we added most of the results on SMOS L2 and SMOS L3 as supplementary information. We will leave that information as supplement. However, Table S1 will be improved with information on single scattering albedo and roughness parameters for the three approaches as recommended by reviewer #1 and by reviewer #2, here below.

In addition, lines 2-8 of page 4 will be moved above Sect. 2.1.1 to strength that those details are common to the three algorithm and so to focus Sects 2.1.1-2.1.3 on the differences.

Furthermore, the presentation of the results for different SMOS L-VOD data sets in Sect. 4.1 will be developed adding the discussion on Spearman and Kendall rank correlation values as follows:

*The rank correlation values with respect to all the evaluation datasets are also higher for IC L-VOD (rho 0.78-0.91, tau 0.61-0.75), followed by L2 L-VOD (rho 0.67-0.83, tau 0.50-0.65) and L3 L-VOD (rho 0.66-0.80, tau 0.49-0.62). These results are in agreement with those obtained with the Pearson correlation and imply that the lower Pearson correlation values obtained for the L3 and L2 datasets are not due to a correlation that could be better but more non-linear than that of the IC dataset. Therefore, using eight vegetation-related evaluation data sets and three different metrics, the most consistent SMOS L-VOD data set is SMOS-IC. This result implies*

*that, currently, the SMOS-IC dataset is the best SMOS L-VOD product to perform vegetation studies, and the rest of the current study will focus on SMOS-IC L-VOD.*

We hope that a potentially interested reader asking himself what is the product to adopt between the L2, L3 and SMOS IC for vegetation monitoring would have a clearer statement to make a choice.

**The relevance of L-VOD for AGB monitoring** is addressed in the rest of Sect 4 and in the new Sect. 5 proposed to discuss the new Fig. 4 as explained above.

Despite these general issues I believe that the topic is relevant, the results are obtained with a sounding scientific approach and the supporting figures and tables are clear. Therefore, I would recommend the paper for publication after a careful revision process.

We thank the reviewer for these encouraging comments.

Specific comments:

In the section 2, "Data", is introduced the SMOS mission and the three different algorithms, considered to retrieve the L band VOD from the SMOS brightness temperature. At line 28 of page 2 is stated that only ascending orbits are considered in the study but the declaimed better overall quality of ascending pass acquisitions appears not justified. Therefore, the authors should provide a better explanation about this important constrain.

It is not really a constrain. The results do not depend significantly on the year or the type or orbits used as reference. The shape of the scatter plots is very similar using data for next years. See for instance the next figure and compare to Fig S3.

[Figure]

[Figure]

Fig. Left : 2012 Ascending orbits. Right: 2012 Descending orbits

The next table shows the parameters of the fits using ascending or descending orbits in 2012. The values can be compared to those of Table S2. They have almost the same numeric values.

| AGB | Year | Orbit | curve | $a$ | $b$ | $c$ | $d$ | $R^2$ |
|---|---|---|---|---|---|---|---|---|
| Avitabile | 2012 | A | 05th | 262.697 | 14.406 | 0.838 | 5.347 | 0.99548 |
| Avitabile | 2012 | A | Mean | 370.615 | 8.920 | 0.727 | 4.915 | 0.99917 |
| Avitabile | 2012 | A | 95th | 462.962 | 9.398 | 0.580 | 1.613 | 0.99243 |
| Saatchi | 2012 | A | 05th | 333.496 | 4.642 | 0.906 | -3.338 | 0.99080 |
| Saatchi | 2012 | A | Mean | 280.675 | 6.686 | 0.684 | 14.439 | 0.99334 |
| Saatchi | 2012 | A | 95th | 291.444 | 9.661 | 0.545 | 32.436 | 0.99372 |
| Baccini | 2012 | A | 05th | 497.748 | 2.673 | 1.024 | -41.748 | 0.98810 |
| Baccini | 2012 | A | Mean | 414.033 | 3.503 | 0.717 | -27.346 | 0.99882 |
| Baccini | 2012 | A | 95th | 383.823 | 4.950 | 0.550 | -1.911 | 0.99757 |
| Bouvet-Mermoz | 2012 | A | 05th | 292.291 | 4.800 | 0.963 | 4.717 | 0.97445 |
| Bouvet-Mermoz | 2012 | A | Mean | 319.123 | 5.261 | 0.760 | 7.879 | 0.99616 |
| Bouvet-Mermoz | 2012 | A | 95th | 358.172 | 7.290 | 0.581 | 17.621 | 0.99226 |
| Avitabile | 2012 | D | 05th | 264.867 | 13.984 | 0.833 | 5.353 | 0.99548 |
| Avitabile | 2012 | D | Mean | 368.991 | 9.271 | 0.726 | 5.795 | 0.99765 |
| Avitabile | 2012 | D | 95th | 483.476 | 7.893 | 0.614 | -2.245 | 0.99585 |
| Saatchi | 2012 | D | 05th | 332.540 | 4.976 | 0.890 | -5.476 | 0.99111 |
| Saatchi | 2012 | D | Mean | 287.547 | 6.436 | 0.691 | 12.981 | 0.99499 |
| Saatchi | 2012 | D | 95th | 291.493 | 8.979 | 0.543 | 30.449 | 0.99324 |
| Baccini | 2012 | D | 05th | 606.345 | 2.250 | 1.168 | -53.195 | 0.98742 |
| Baccini | 2012 | D | Mean | 398.354 | 3.688 | 0.706 | -21.945 | 0.99806 |
| Baccini | 2012 | D | 95th | 379.008 | 5.076 | 0.567 | 6.162 | 0.99538 |
| Bouvet-Mermoz | 2012 | D | 05th | 308.430 | 4.441 | 0.965 | 0.250 | 0.98165 |
| Bouvet-Mermoz | 2012 | D | Mean | 317.532 | 5.425 | 0.764 | 10.022 | 0.99729 |
| Bouvet-Mermoz | 2012 | D | 95th | 375.791 | 6.420 | 0.604 | 12.619 | 0.99451 |

Therefore, there is no real impact of using just one year or using ascending or descending orbits. However, in a revised version of the manuscript we will use two years of data both for ascending and descending orbits. The results will not change but they would look more robust to the reader. This will be discussed in a revised version of the manuscript. See also answer to General Comment 3.

In the following subsection are introduced the ESA L2 algorithm, the CATDS L3 algorithm and the INRA-CESBIO algorithm that were applied to obtain three different L band VOD data sets.  If the Authors are inclined to stress the intercomparison between the outcomes of the different retrieval approach, a deeper discussion about the different algorithm could be effective to introduce the subsequent results, i.e. figure 1 and table 1.  This choice, could be a good solution to solve some ambiguities between the study aim, as claimed on the paper title, and the interesting overview of the different algorithms performances.  Anyway, a better explanation on the assumptions (i.e.  soil roughness and albedo) under which the three different algorithms are based should be provided.

Thanks for pointing this out. This point has also been raised by referee #1.

In a revised version, first, the albedo will be cites earlier since the introduction citing presenting the tau-omega model. Therefore, the lines 10-14 of page 2 :

*In the presence of vegetation, part of the soil emission is absorbed and scattered. This extinction effect is parameterized by the vegetation optical depth (VOD) that can be estimated using radiative transfer theory […] Wigneron et al. 2007).*

It will be rephrased to:

*In the presence of vegetation, part of the soil emission is absorbed and scattered. These effects can be parameterized using radiative transfer models such as the so-called tau-omega model (Refs), were tau is the optical depth and omega is the single scattering albedo. Tau was shown to be linked […] Wigneron et al. 2007). Therefore, tau is commonly known as Vegetation Optical Depth (VOD).*

Regarding the actual values used for different SMOS product, the information will be added more clearly in Sect. 2 and in Table S1. For SMOS IC the roughness and single scattering parameters are assigned per IBGP classes, based on Parrens et al. (2017a, b), and are averaged per pixel according to the fraction of classes present in the pixel (Fernandez-Moran et al. 2017).

For SMOS L2 and L3 algorithms, single scattering albedo and roughness values depend on the surface type and are taken from literature and/or specific SMOS studies. For low vegetated area the single scattering albedo is set to 0 and roughness set to 0.1. For forested areas the single scattering albedo is set to 0.06 for tropical and subtropical forest and 0.08 for Boreal forest and roughness set to 0.3 (Rahmoune et al.  2013,2014, Al Bitar et al. 2017).

Marie Parrens, Jean-Pierre Wigneron, Philippe Richaume, Ahmad Al Bitar, Arnaud Mialon, Roberto Fernandez-Moran, Amen Al-Yaari, Peggy O'Neill, and Yann Kerr, 2017. Considering Combined or Separated Roughness and Vegetation Effects in Soil Moisture Retrievals, International Journal of Applied Earth Observation and Geoinformation 55, 73-86.

*M. Parrens, A. Al Bitar, A. Mialon, R. Fernandez-Moran, J.-P. Wigneron, P. Ferrazzoli, and Y. Kerr, 2017. Estimation of the L-band Effective Scattering Albedo of Tropical Forests using SMOS observations, IEEE GRS Letters 2017, 14, 1223-1227*

After the introduction of the VOD datasets the different benchmark sets are presented.  In the section 2.2.1 it is introduced the Worldclim data set, that is used to infer the relationship between the L band VOD and the mean annual precipitation.  This analysis seems meaningless since, as it is reported at line 15 of page 5, the considered precipitation is extracted from a dataset ranging only between "1970-2000". This point should be clarified also considering that the relationship between the precipitation and the VOD are not well commented in the paper.

The regime of precipitations can be a driver of the vegetation conditions. In a revised version, precipitations will be discussed independently of the other datasets. First, Fig S6 will be moved to the main documents and all IGBP classes will be discussed independently instead of grouping some of them. Panels c and d of figure 3 will be now a single new figure. The rest of the panels of Fig. 3 will be removed as they will be redundant with those of the new figures showed above. Here, showing only two groups of biomes is pertinent as we want, as the reviewer says, to show the possible link of precipitation as one of the drivers of the vegetation properties. Thus, we show that there are basically two regimes. In the first one, as precipitation increases the amount of vegetation as traced by AGB maps, VOD or NDVI/EVI increases. In contrast, there is threshold of ~1500 mm/year over which AGB, VOD and NDVI are decoupled from the amount of the annual precipitations.

Taking this into account, lines 13-17 of page 10 will be replaced by :

*Regarding the relationship with respect to annual precipitation, Fig. 7 shows the precipitations and L-VOD scatter plots for two land cover groups: (i) grasslands, croplands, shrublands, savannahs and woody savannahs and (ii) ever- green broadleaf forest. For the first group, L-VOD increases with increasing annual precipitations until a value of ~ 0.7 for ~ 1500 mm. In this range of L-VOD all other vegetation tracers increase as well. For instance, Bouvet-Mermoz and Saatchi's AGB increase up to 85 Mg/h and ~ 100 Mg/h, respectively, and NDVI and DVI increase up to ~ 0.7 and ~ 0.45, respectively (Fig. 6). In contrast, over that threshold of ~ 1500 mm of annual precipitations, which occurs basically in the evergreen broadleaf forest, L-VOD and the other vegetation tracers are not linked to the amount of precipitation.*

In the section 2.2.4 are presented the different AGB datasets considered as benchmarks.  Here the sentence "This study used four static AGB benchmark maps (Baccini et al., 2012; Saatchi et al.,  2011;  Avitabile et al.,  2016;  Bouvet et al., 2018) each with specific strengths and limitations to assess L-VOD's ability to reflect aboveground biomass in different" is questionable and not well supported by the results.  In particular, the Avitabile dataset is obtained by the fusion of the Baccini and Saatchi maps through a machine learning approach and it is proved to outperform the previous datasets in terms of retrieved AGB accuracy.  The Authors should argue better the aspects related to the analysis carried out with these three different AGB data sets.  On the contrary the consideration of the Bouvet dataset is very interesting and should be emphasized.

We are afraid we disagree. We do not see clear evidence that the Avitabile "outperforms" any other AGB dataset. We do think that AGB estimation from remote sensing measurements is complex and the errors of different retrieval methods are not easy to characterize. The fact that the Avitabile dataset is so different to both the Baccini and Saatchi maps used as input is actually puzzling, for instance the sharp decrease AGB from the Equatorial region with distance is not seen in any

other AGB map. The Avitabile method can be biased to high AGB values because most of the plots used as reference are in dense forest. Furthermore, a totally independent observable such as L-VOD shows clear relationships for low AGB for all the datasets but Avitabile.

Otherwise, the analysis is performed in exactly the same way for all AGB datasets and the more considerations on Bouvet dataset are proposed to be more taken into account by moving Fig. S6 to the main manuscript and discussing in more details the results for different biomes:

The best correlations of AGB and L-VOD are found with (i) Bouvet-Mermoz for Shrublands and Savannahs (ii) Baccini for croplands and equatorial forest (iii) Saatchi for grasslands. Regarding natural vegetation and woody savannah the correlation values obtained with Saatchi and Baccini are very similar. One should note that correlation values obtained with Bouvet-Mermoz for woody savannah are degraded to the fact that for the highest values of AGB found in this class, at the SMOS resolution, the AGB estimation is a mix of Bouvet and Mermoz approaches. Therefore, all AGB datasets except that of Avitabile performs the best for a few land cover classes.

Interpreting where do this differences come from is not easy. Radar observations in low vegetation regions such as shrublands and grasslands are thought to be very sensitive to biomass variations, in spite of a significant sensitivity to soil moisture. The high correlation of the two AGB maps mainly based of radar data (Saatchi and Bouvet-Mermoz) with SMOS L-VOD in grasslands would confirm this fact, as the high correlation in shrublands for Bouvet-Mermoz.

 The scatter plot found with Avitabile for woody-savannah resembles an overlay of the scatter plot obtained with Baccini and the scatter plot obtained with Saatchi. The low AGB vs L-VOD slopes obtained for low vegetation classes, significantly lower than those found with the original Saatchi and Baccini datasets, are rather difficult to understand. The strange behavior of Avitabile AGB probably comes from the fact that it is pure data driven method and that it is therefore very sensitive to the data used to train the method. In their training database, high AGB plots could be over-represented.

In addition as mentioned above, the distribution of Baccini AGB for woody savannah is significantly different to the other datasets, which much higher values.

These elements will be added to the results section when discussing the new figures that will replace Fig. 3 and S6.

 In the Results section it should be provided a deeper explanation of the research outcomes, in particular the scatterplots reported in figure 2 need to be better commented.

We agree. We have realized from comments by Referees #1 and #2 that the explanation of the results should be improved. Therefore, as already mention in the answer to the previous comment, Fig. S6 will be moved to the main manuscript and more details on the results for different biomes will be given as proposed in the answer to the previous comment. This will make clear than the relationships shown in Fig. 2 are actually a composition of different relationships for different biomes. See below the new figures:

[Figure]

**Figure 4.** SMOS IC L-VOD relationships to the AGB and tree heigh evaluation datasets for different land cover classes: From left to right:
Open shrublands. *(ii)* Croplands, *(iii)* Grasslands, and *(iv)* Cropland/natural vegetation mosaics. See Table S2 for more details on these
nd cover classes.

[Figure]

**Figure 5.** SMOS IC L-VOD relationships to the AGB and tree heigh evaluation datasets for different land cover classes: From left to right: *(i)* Savannah *(iii)* Woody savannah and *(iv)* Evergreen broadleaf. See Table S2 for more details on these land cover classes.